# Unleashing the Potential of Acquisition Functions in High-Dimensional Bayesian Optimization

**An empirical study to understand the role of acquisition function maximizer initialization**

**Jiayu Zhao**  *scjzh@leeds.ac.uk*
*School of Computing*
*University of Leeds*

**Renyu Yang**  *r.yang1@leeds.ac.uk*
*School of Computing*
*University of Leeds*

**Shenghao Qiu**  *sc19sq@leeds.ac.uk*
*School of Computing*
*University of Leeds*

**Zheng Wang**  *z.wang5@leeds.ac.uk*
*School of Computing*
*University of Leeds*

**Reviewed on OpenReview:** *https://openreview.net/forum?id=0CM7Hfsy61*

## Abstract

Bayesian optimization (BO) is widely used to optimize expensive-to-evaluate black-box functions. BO first builds a surrogate model to represent the objective function and assesses its uncertainty. It then decides where to sample by maximizing an acquisition function (AF) based on the surrogate model. However, when dealing with high-dimensional problems, finding the global maximum of the AF becomes increasingly challenging. In such cases, the initialization of the AF maximizer plays a pivotal role, as an inadequate setup can severely hinder the effectiveness of the AF.

This paper investigates a largely understudied problem concerning the impact of AF maximizer initialization on exploiting AFs' capability. Our large-scale empirical study shows that the widely used random initialization strategy often fails to harness the potential of an AF. In light of this, we propose a better initialization approach by employing multiple heuristic optimizers to leverage the historical data of black-box optimization to generate initial points for the AF maximizer. We evaluate our approach with a range of heavily studied synthetic functions and real-world applications. Experimental results show that our techniques, while simple, can significantly enhance the standard BO and outperform state-of-the-art methods by a large margin in most test cases.

## 1 Introduction

Bayesian optimization (BO) is a well-established technique for expensive black-box function optimization. It has been used in a wide range of tasks - from hyper-parameter tuning (Bergstra et al., 2011), onto chemical material discovery (Hernández-Lobato et al., 2017) and robot control and planning (Lizotte et al., 2007; Martinez-Cantin et al., 2009). BO tries to improve sampling efficiency by fitting a probabilistic surrogate model (usually a Gaussian process (Seeger, 2004)) to guide its search. This model is used to define an acquisition function (AF) that trades off exploitation (model prediction) and exploration (model uncertainty). Maximizing the AF will get the next sequential query point that BO thinks is promising.

While BO shows good performance for low-dimensional problems, its application to high-dimensional problems is often not competitive with other techniques (Eriksson et al., 2019). Given that BO's performance depends on both the model-based AF itself and the process of maximizing the AF, either of them can be a bottleneck for high-dimensional BO (HDBO). The vast majority of the prior work in BO has focused on the former, i.e., designing the surrogate model and AF (Snoek et al., 2012; Srinivas et al., 2009; Wu & Frazier, 2016; Wang & Jegelka, 2017; Oh et al., 2018; Moss et al., 2021; Snoek et al., 2014). Little work is specialized for improving the latter. A recent development in maximizing AF implements a multi-start gradient-based AF maximizer in batch BO scenarios, achieving better AF maximization results than random sampling and evolutionary algorithms (Wilson et al., 2018). However, as the dimensionality increases, even the multi-start gradient-based AF maximizer struggles to globally optimize the AF. In such cases, the initialization of the AF maximizer greatly influences the quality of AF optimization. Yet, it remains unclear how AF maximizer initialization may impact the utilization of AF's potential and the end-to-end BO performance. Upon examining the implementations of the state-of-the-art BO packages, we found that random initialization (selecting initial points from a set of random points) or a variant is a typical default strategy for initializing the AF maximizer. This is the case for widely-used BO packages like BoTorch (Balandat et al., 2020), Skopt (Head et al., 2021), Trieste (Picheny et al., 2023), Dragonfly (Kandasamy et al., 2020) and GPflowOpt (Knudde et al., 2017a). Specifically, GPflowOpt, Trieste and Dragonfly select the top $n$ points with the highest AF values from a set of random points to serve as the initial points. BoTorch uses a similar but more exploratory strategy by performing Boltzmann sampling rather than top-n selection on random points. Likewise, Skopt directly selects initial points by uniformly sampling from the global search domain. Furthermore, various HDBO implementations adopt random initialization as their default strategy (Oh et al., 2018; Letham et al., 2020; Wang et al., 2018; Kandasamy et al., 2015; Wu & Frazier, 2016; Wang et al., 2017). GPyOpt (The GPyOpt authors, 2016) and Spearmint (Snoek et al., 2012) are two of the few works that provide different initialization strategies from random initialization. GPyOpt combines random points with the best-observed points to serve as the initial points. Spearmint uses a Gaussian spray around the incumbent best to generate initial points. However, no empirical study shows that these initialization strategies are better than random initialization. As such, random initialization remains the most popular strategy for HDBO AF maximization. There is a need to understand the role of the initialization phase in the AF maximization process.

The paper systematically studies the impact of AF maximizer initialization in HDBO. This is motivated by an observation that the pool of available candidates generated during AF maximization often limits the AF's power when employing the widely used random initialization for the AF maximizer. This limitation asks for a better strategy for AF maximizer initialization. To this end, our work provides a simple yet effective AF maximizer initialization method to unleash the potential of an AF. Our goal is not to optimize an arbitrary AF but to find ways to maximize the potential of reasonable AF settings. Our key insight is that when the AF is effective, the historical data of black-box optimization could help identify areas that exhibit better black-box function values and higher AF values than those obtained through random searches of AF.

We develop AIBO[1], a Python framework to employ multiple heuristic optimizers, like the covariance matrix adaptation evolution strategy (CMA-ES) (Hansen et al., 2003) and genetic algorithms (GA) (Alam et al., 2020), to utilize the historical data of black-box optimization to generate initial points for a further AF maximizer. We stress that the heuristics employed by AIBO are not used to optimize the AF. Instead, they capitalize on the knowledge acquired from the already evaluated samples to provide initial points to help an AF maximizer find candidate points with higher AF values. For instance, CMA-ES generates candidates from a multivariate normal distribution determined by the historical data of black-box optimization. To investigate whether performance gains come from better AF maximization, AIBO also incorporates random initialization for comparison. Each BO iteration runs multiple AF initialization strategies, including random initialization on the AF maximizer, to generate multiple candidate samples. It then selects the sample with the maximal AF value for black-box evaluation. Thus, heuristic initialization strategies work only when they identify higher AF values than random initialization.

To demonstrate the benefit of AIBO in black-box function optimizations, we integrate it with the multi-start gradient-based AF maximizer and apply the integrated system to synthetic test functions and real-world applications with a search dimensionality ranging between 14 and 300. Experimental results show that

---

[1]AIBO =Acquisition function maximizer Initialization for Bayesian Optimization.

AIBO significantly improves the standard BO under various AF settings. Our analysis suggests that the performance improvement comes from better AF maximization, highlighting the importance of AF maximizer initialization in unlocking the potential of AF for HDBO.

The contribution of this paper is two-fold. Firstly, it investigates a largely ignored yet significant problem in HDBO concerning the impact of the initialization of the AF maximizer on the realisation of the AF capability. It empirically shows the commonly used random initialization strategy limits AFs' power, leading to over-exploration and poor HDBO performance. Secondly, it proposes a simple yet effective initialization method for maximizing the AF, significantly improving the performance of HDBO. We hope our findings can encourage more research efforts in optimizing the initialization of AF maximizers of HDBO.

**Data availability**  The data and code associated with this paper are openly available at `https://github.com/gloaming2dawn/AIBO`.

## 2 Related Work

### 2.1 High-dimensional Bayesian Optimization

Prior works in HDBO have primarily focused on dimensionality reduction or pinpointing the performance bottleneck. There are two common approaches for dimensionality reduction. The first assumes the black-box function has redundant dimensions. By mapping the high-dimensional space to a low-dimensional subspace, standard BO can be done in this low-dimensional space and then projected up to the original space for function evaluations (Wang et al., 2013; Letham et al., 2020; Binois et al., 2020; Qian et al., 2016). A second approach targets functions with additive structures, i.e., cases where variables in the design space are separable (Kandasamy et al., 2015; Wang et al., 2017; Gardner et al., 2017; Rolland et al., 2018; Li et al., 2018). Especially, LineBO (Kirschner et al., 2019) restricts the optimization problem to a sequence of iteratively chosen one-dimensional sub-problems. Both strategies are inadequate for many real-life scenarios where the black-box function does not exhibit additive structures or lacks redundancy in dimensions. Besides these dimensionality reduction techniques, efforts have been made to improve high-dimensional BO directly (Wang et al., 2018; Rana et al., 2017; Oh et al., 2018; Eriksson et al., 2019), with TuRBO (Eriksson et al., 2019) as the state-of-the-art method.

### 2.2 Acquisition Function Maximization

Given the posterior belief, BO uses an AF to select new queries. Random sampling, evolutionary algorithms and gradient-based optimization are three mainstreamed AF maximization techniques. Random sampling is efficient in low-dimensional problems (Bartz-Beielstein et al., 2005; Hutter et al., 2009; 2010) but can be inadequate for high-dimensional problems (Hutter et al., 2011). Evolutionary algorithms are often used where gradient information is unavailable (Kandasamy et al., 2020; Cowen-Rivers et al., 2020). For AFs that support gradient information, a multi-start gradient-based optimization method is a good choice for AF optimization (Wilson et al., 2018). Our end-to-end BO framework thus builds upon this technique.

Despite the importance of AF maximization, little attention has been paid to optimizing the initialization phase for AF maximizers. Most prior work (Snoek et al., 2012; Knudde et al., 2017b; Klein et al., 2017; Wu & Frazier, 2016; Kandasamy et al., 2020; Balandat et al., 2020; Oh et al., 2018; Kandasamy et al., 2015; Wang et al., 2018; Cowen-Rivers et al., 2020; Letham et al., 2020; Nayebi et al., 2019) uses random initialization. This simple strategy can be effective in low dimensions, but as we will show in the paper, it is ill-suited for high-dimensional problems. SMAC (Hutter et al., 2011) and Spearmint (Snoek et al., 2012) are a few BO techniques that do not use random initialization. Instead, they use a Gaussian spray around the incumbent best to generate initial points to initialize its local maximizer. Our work provides a systematic study to empirically demonstrate the importance of initialization for AF maximization. It proposes an initialization optimization to improve prior work by leveraging multiple heuristic algorithms to more effectively utilize the evaluated samples, significantly improving the performance of a given AF.

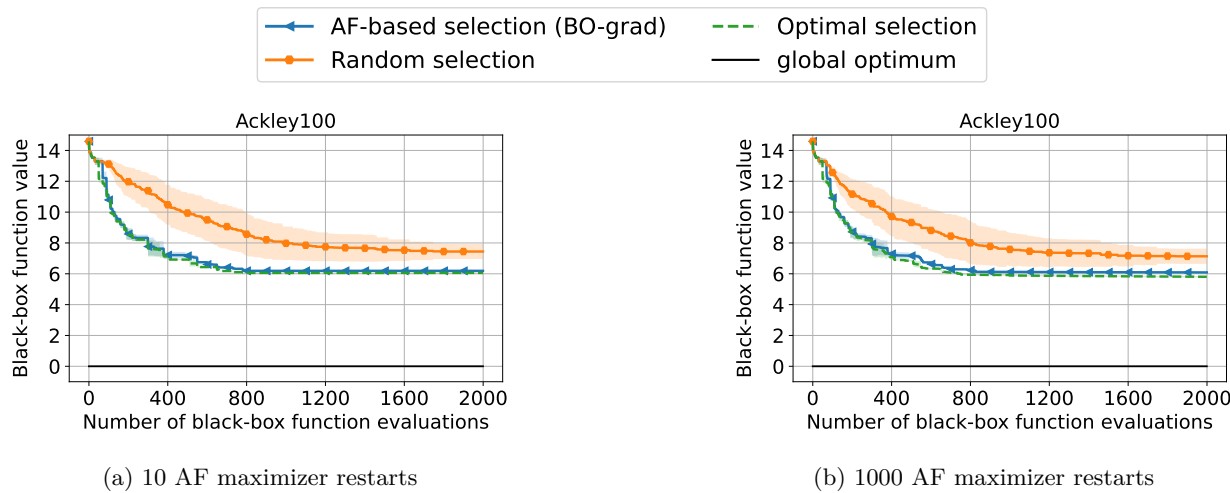

(a) 10 AF maximizer restarts

(b) 1000 AF maximizer restarts

Figure 1: Evaluating the sample chosen by AF-based selection against random and optimal selection among all intermediate candidate points generated during the AF maximization process when applying BO-grad to 100D Ackley functions. We use two random initialization settings for AF maximization: **(a)** 10 restarts and **(b)** 1000 restarts. In both settings, the performance of the native BO-grad (AF-based selection) is close to optimal selection and better than random selection, suggesting that the AF is effective at selecting a good sample from all candidates but is restricted by the pool of available candidates. Increasing the number of restarts from 10 to 1000 does not enhance the quality of intermediate candidates, indicating that a more effective initialization scheme, as opposed to random initialization, is necessary.

## 3 Motivation

As a motivation example, consider applying BO to optimize a 100-dimensional black-box Ackley function that is extensively used for testing optimization algorithms. The goal is to find a set of input variables ($x$) to minimize the output, $f(x_1, \dots, x_{100})$. The search domain is $-5 \leq x_i \leq 10, i = 1, 2, \dots, 100$, with a global minimum of 0. For this example, we use a standard BO implementation with a prevalent AF[2], Upper Confidence Bound (UCB) (Srinivas et al., 2009), denoted as:

$$\alpha(x) = -\mu(x) + \sqrt{\beta_t} \cdot \sigma(x) \tag{1}$$

where $\mu(x)$ and $\sigma(x)$ are the posterior mean (prediction) and posterior standard deviation (uncertainty) at point $x$ predicted by the surrogate model, and $\beta_t$ is a hyperparameter that trades off between exploration and exploitation. we set $\beta_t = 1.96$ in this example.

Here, we use random search to create the initial starting points for a *multi-start gradient-based* AF maximizer to iteratively generate multiple candidates, from which the AF chooses a sample for evaluation. In each BO iteration, we first evaluate the AF on 100000 random points and then select the top $n$ points as the initial points for the further gradient-based AF maximizer. We denote this implementation as `BO-grad`. In this example, we use two settings $n = 10$ and $n = 1000$.

As shown in Fig. 1(a), the function output given by BO-grad with 10 AF maximizer restarts is far from the global minimum of 0. We hypothesize that while the AF is effective, the low quality of candidate samples generated in AF maximization limits the power of the AF. To verify our hypothesis, we further consider two strategies: (1) either randomly select the next query point from all the candidate points generated during AF maximization or (2) exhaustively evaluate them at each BO iteration. The former and latter strategies correspond to "**random selection**" and "**optimal selection**" schemes, respectively. Despite the ideal but costly "optimal selection" search scenario, BO does not converge well, indicating intrinsic deficiencies in the AF maximization process. Meanwhile, the AF itself can choose a good candidate sample point to evaluate,

---

[2]Other acquisition functions include the probability of improvement (PI) and expected improvement (EI).

as the performance of the native AF-based BO-grad is close to that of "optimal selection" and better than that of "random selection". This observation suggests that the AF is effective at selecting a good sample in this case but its power is severely limited by the candidate samples generated during the AF maximization process. We also observe similar results manifest when using other representative maximizers like random sampling and evolutionary algorithms.

We then test what happens when we increase the number of AF maximization restarts of BO-grad to generate more candidates for AF to select at each iteration. However, in Fig. 1(b), it is evident that even with an increase in random restarts to 1000, the quality of intermediate candidate points generated during the AF maximization process remains similar to that with 10 restarts. Furthermore, in the case of 1,000 restarts, the performance of BO-grad is still close to that of "optimal selection", reinforcing our observation that the pool of candidates restricts AF's power. This observation suggests that we need a better initialization scheme rather than simply increasing the number of restarts of random initialization.

This example motivates us to explore the possibility of improving the BO performance by providing the AF with better candidate samples through enhanced AF maximizer initialization. Our intuition is that the potential of AF is often not fully explored in HDBO. Moreover, the commonly used random initialization of the AF maximization process is often responsible for inferior candidates. We aim to improve the quality of the suggested samples through an enhanced mechanism for AF maximizer initialization. As we will show in Section 6, our strategy significantly improves BO on the 100D Ackley function, finding an output minimum of less than 0.5, compared to 6 given by BO-grad after evaluating 5,000 samples.

## 4    Methodology

Our study focuses on evaluating the initialization phase of AF maximization. To this end, we developed AIBO, an open-source framework to facilitate an exhaustive and reproducible assessment of AF maximizer initialization methods.

### 4.1    Heuristic Acquisition Function Maximizer Initialization

AIBO leverages multiple heuristic optimizers' candidate generation mechanisms to generate high-quality initial points from the already evaluated samples. Given the proven effectiveness of heuristic algorithms in various black-box optimization scenarios, they are more likely to create initial candidates near promising regions. As an empirical study, we aim to explore whether this initialization makes the AF optimizer yield points with higher AF values and superior black-box function values compared to random initialization.

As described in Algorithm 1, AIBO maintains multiple black-box heuristic optimizers $o_0, o_2, ...o_{l-1}$. At each BO iteration, each heuristic optimizer $o_i$ is asked to generate $k$ raw points $X^i$ based on its candidate generation mechanisms (e.g., CMA-ES generates candidates from a multivariate normal distribution). AIBO then selects the best $n$ points $\widetilde{X}^i$ from $X^i$ for each optimizer $o_i$, respectively. After using these points to initialize and run an AF maximizer for each initialization strategy, we obtain multiple candidates $x_t^0, x_t^1, ..., x_t^{l-1}$. Finally, the candidate with the highest AF value is chosen as the sample to be evaluated by querying the black-box function. Crucially, the evaluated sample is used as feedback to update each optimizer $o_i$ - for example, updating CMA-ES's normal distribution. This process repeats at each subsequent BO iteration.

Our current default implementation employs CMA-ES, GA and random search as heuristics for initialization. We use the "combine-then-select" approach because it allows us to examine if GA/CMA-ES initialization could find better AF values than random initialization. Our scheme only chooses GA/CMA-ES initialization if it yields larger AF values than random initialization. Besides, while heuristics like GA already provide exploratory mechanisms and altering its hyperparameters can achieve different trade-offs, the usage of random initialization here could also mitigate the case of over-exploitation.

**CMA-ES**    CMA-ES uses a multivariate normal distribution $\mathcal{N}(m, C)$ to generate initial candidates in each BO iteration. Here, the mean vector $m$ determines the center of the sampling region, and the covariance matrix $C$ determines the shape of the region. The covariance matrix $m$ is initialized at the beginning of the BO search, and each direction (dimension) will be assigned an initial covariance, ensuring exploration across

---

**Algorithm 1** Acquisition function maximizer initialization for high-dimensional Bayesian optimization (AIBO)

---

    **Input**: The number of search iterations $T$
    **Output**: The best-performing query point $x^*$
1: Draw $N$ samples uniformly to obtain an initial dataset $D_0$
2: Specify a set of heuristic optimizers $\mathcal{O}$, where the size is $l$
3: Use $D_0$ to initialize a set of heuristic optimizers $\mathcal{O}$
4: **for** $t = 0 : T - 1$ **do**
5:     Fit a Gaussian process $\mathcal{G}$ to the current dataset $D_t$
6:     Construct an acquisition function $\alpha(x)$ based on $\mathcal{G}$
7:     **for** $i = 0 : l - 1$ **do**
8:         $\mathbf{X}^i \leftarrow o_i.ask(num = k)$                                   $\triangleright$ Ask the heuristic to generate $k$ candidates
9:         $\widetilde{\mathbf{X}}^i \leftarrow top(\alpha(\mathbf{X}^i), n)$               $\triangleright$ Select top-n ($n < k$) candidates from $\mathbf{X}^i$ according to $\alpha(x)$
10:        Use $\widetilde{\mathbf{X}}^i$ to initialize an acquisition function maximizer $\mathcal{M}$
11:        $x_t^i \leftarrow \underset{x \in \mathcal{X}}{\arg\max}\, \alpha(x) | \mathcal{M}$                                      $\triangleright$ Use $\mathcal{M}$ to maximize $\alpha(x)$
12:     **end for**
13:     $x_t \leftarrow \arg\max \alpha(x)\ x \in \{x_t^0, x_t^1, ..., x_t^{l-1}\}$              $\triangleright$ Select the point with the highest AF value
14:     $y_t \leftarrow f(x_t)$                                                  $\triangleright$ Evaluate the selected sample
15:     **for each** $o_i \in \mathcal{O}$ **do**
16:        $o_i.tell(x_t, y_t)$                                     $\triangleright$ Update heuristic optimizer $o_i$ with $(x_t, y_t)$
17:     **end for**
18:     Update dataset $D_{t+1} = D_t \cup \{(x_t, y_t)\}$
19: **end for**

---

all directions. By updating $m$ and $C$ using new samples after each BO iteration, CMA-ES can gradually focus on promising regions.

**GA**   GA keeps a population of samples to determine its search region. It uses biologically inspired operators like mutation and crossover to generate new candidates based on the current population. Its population is updated by newly evaluated samples after each BO iteration.

**Random**   Most BO algorithms or library implementations use random search for initializing the AF maximizer. We use it here to eliminate the possibility of AIBO's performance improvement stemming from GA/CMA-ES initialization, yielding points with better black-box function values but smaller AF values.

Our heuristic initialization process is AF-related, as the heuristic optimizers are updated by AF-chosen samples. Usually, a more explorative AF will make the heuristic initialization also more explorative. For instance, in GA, if the AF formula leans towards exploration, the GA population composed of samples chosen by this AF will have greater diversity, leading to generating more diverse raw candidates. The details of how GA and CMA-ES generate candidates and update themselves are provided in Appendix A.2.

## 4.2 Implementation Details

Since this study focuses on the AF maximization process, we utilize other BO settings that have demonstrated good performance in prior work. We describe the implementation details as follows.

**Gaussian process regression**   To support scalable GP regression, we implement the GP model based on an optimized GP library GPyTorch (Gardner et al., 2018). GPyTorch implements the GP inference via a modified batched version of the conjugate gradients algorithm, reducing the asymptotic complexity of exact GP inference from $O(n^3)$ to $O(n^2)$. The overhead of running BO with a GP model for a few thousand evaluations should be acceptable for many scenarios that require hundreds of thousands or more evaluation iterations.

We select the Matérn-5/2 kernel with ARD (each input dimension has a separate length scale) and a constant mean function to parameterize our GP model. The model parameters are fitted by optimizing the log-marginal likelihood before proposing a new batch of samples for evaluation. Following the usual GP fitting procedure, we re-scale the input domain to $[0, 1]^d$. We also use power transforms to the function values to

Table 1: Benchmarks used in evaluation.

|  | Function/Task | #Dimensions | Search Range |
|---|---|---|---|
| **Synthetic** | Ackley | 20, 100, 300 | [-5, 10] |
|  | Rosenbrock | 20, 100, 300 | [-5, 10] |
|  | Rastrigin | 20, 100, 300 | [-5.12, 5.12] |
|  | Griewank | 20, 100, 300 | [-10, 10] |
|  | Levy | 20, 100, 300 | [-600, 600] |
| **Robotics** | Robot pushing | 14 | / |
|  | Rover trajectory planning | 60 | [0, 1] |
|  | Half-Cheetah locomotion | 102 | [-1, 1] |

make data more Gaussian-like. This transformation is useful for highly skewed functions like Rosenbrock and has been proven effective in real-world applications (Cowen-Rivers et al., 2020). We use the following bounds for the model parameters: length-scale $\lambda_i \in [0.005, 20.0]$, noise variance $\sigma^2 \in [1e^{-6}, 0.01]$.

**Batch Bayesian optimization** To support batch evaluation for high-dimensional problems, we employ the UCB and EI AFs estimated via Monte Carlo (MC) integration. Wilson et al. (2018) have shown that MC AFs naturally support queries in parallel and can be maximized via a greedy sequential method. Algorithm 1 shows the case where the batch size is one. Assuming the batch size is $q$, the process of greedy sequential acquisition function maximization can be expressed as follows:

1. Maximize the initial MC acquisition function $\alpha^0(x)$ to obtain the first query point $x_0$.
2. Use the first query sample $(x_0, \alpha^0(x_0))$ to update $\alpha^0(x)$ to $\alpha^1(x)$ and maximize $\alpha^1(x)$ to obtain the second query point $x_1$.
3. Similarly, successively update and maximize $\alpha^2(x), \alpha^3(x), ..., \alpha^{q-1}(x)$ and obtain query points $x_2, x_3, ...x_{q-1}$.

We implemented it based on BoTorch (Balandat et al., 2020), which provided the MC-estimated acquisition functions and the interface for function updating via query samples. Details of the calculation of MC-estimated AFs are provided in Appendix A.5.

**Hyper-parameter settings** We use $N = 50$ samples to obtain all benchmarks' initial dataset $D_0$. We set $k = 500$ and $n = 1$ for each AF maximizer initialisation strategy. We use the implementations in pycma (Hansen et al., 2022) and pymoo (Blank & Deb, 2020) for the CMA-ES and the GA initialization strategies, respectively. For CMA-ES, we set the initial standard deviation to 0.2. For GA initialization, we set the population size to 50. The default AF maximizer in AIBO is the gradient-based optimization implemented in BoTorch. The default AF is UCB with $\beta_t = 1.96$ (default setting in the skopt library (Head et al., 2021)), and the default batch size is set to 10. In Section 6.6, we will also show the impact of changing these hyper-parameters in our experiments.

## 5 Experimental Setup

### 5.1 Benchmarks

Table 1 lists the benchmarks and the problem dimensions used in the experiments. These include synthetic functions and three tasks from the robot planning and control domains.

**Synthetic functions** We first apply AIBO and the baselines to four common synthetic functions: Ackley, Rosenbrock, Rastrigin and Griewank. All these functions allow flexible dimensions and have a global minimum of 0. We select 20, 100 and 300 dimensions in our study to show how higher dimensions of the same problem influence the BO performance.

**Robot pushing** The task is used in TurBO Eriksson et al. (2019) and Wang et al. (2018) to validate high-dimensional BOs. The goal is to tune a controller for two robot hands to push two objects to given

target locations. Despite having only 14 dimensions, this task is particularly challenging as the reward is sparse in its search space.

**Rover trajectory planning**   The task, also considered in Eriksson et al. (2019); Wang et al. (2018), is to maximize the trajectory of a rover over rough terrain. The trajectory is determined by fitting a B-spline to 30 points in a 2D plane (thus, the state vector consists of 60 variables). This task's best reward is 5.

**Half-cheetah robot locomotion**   We consider the 102D half-cheetah robot locomotion task simulated in MuJoCo (Todorov et al., 2012) and use the linear policy $a = Ws$ introduced in (Mania et al., 2018) to control the robot walking. Herein, $s$ is the state vector, $a$ is the action vector, and $W$ is the linear policy to be searched for to *maximize* the reward. Each component of $W$ is continuous and within [-1,1].

## 5.2   Evaluation Methodology

We design various experiments to validate the significance of the initialization of the AF maximization process. All experiments are run 50 times for evaluation. In Section 6.1, we evaluate AIBO's end-to-end BO performance by comparing it to various baselines including standard BO implementations with AF maximizer random initialization, heuristic algorithms and representative HDBO methods. In Section 6.2, we evaluate the robustness of AIBO under different AFs. In Section 6.3, we evaluate AIBO's three initialization strategies in terms of AF values, GP posterior mean, and GP posterior variance under different AF settings. This will show whether AIBO's heuristic initialization strategies lead to better AF maximization. In Section 6.4 , we use ablation experiments to examine the impact of the three individual initialization strategies in AIBO. In Section 6.5, we compare AIBO to BO implementations with alternative AF maximizer initialization strategies rather than selecting initial points with the highest AF values from a set of random points. In Section 6.6, we show the impact of hyper-parameters on the performance of AIBO. In Section 6.7, we provide the algorithmic runtime of our method.

## 6   Experimental Results

Highlights of our evaluation results are:

- AIBO significantly improves standard BO and outperforms heuristic algorithms and representative HDBO methods in most test cases (Sec. 6.1 and Sec. 6.2);

- By investigating AIBO's three initialization strategies in terms of AF maximization, we show that random initialization limits AFs' power by yielding lower AF values and larger posterior variances, leading to over-exploration, empirically confirming our hypothesis (Sec. 6.3);

- We provide a detailed ablation study and hyper-parameters analysis to understand the working mechanisms of AIBO (Sec. 6.4 and Sec. 6.6).

### 6.1   Comparison with Baselines

### 6.1.1   Setup

We first compare AIBO to eight baselines: BO-grad, BO-es, BO-random, TuRBO, HeSBO, CMA-ES, GA and AIBO-none. We describe the baselines as follows.

BO-grad, BO-es and BO-random respectively refer to the standard BO implementations using three AF maximizers: multi-start gradient-based, CMA-ES and random sampling. These standard BO implementations use the same base settings as AIBO but with a random initialization scheme for AF maximization. To show the effectiveness of AIBO, BO-grad is allowed to perform more costly AF maximization; we set $k = 2000$ and $n = 10$.

TuRBO (Eriksson et al., 2019), HeSBO (Nayebi et al., 2019) and Elastic BO (Rana et al., 2017) are representive HDBO methods. We use $N = 50$ samples to obtain the initial dataset $D_0$ for all three HDBO

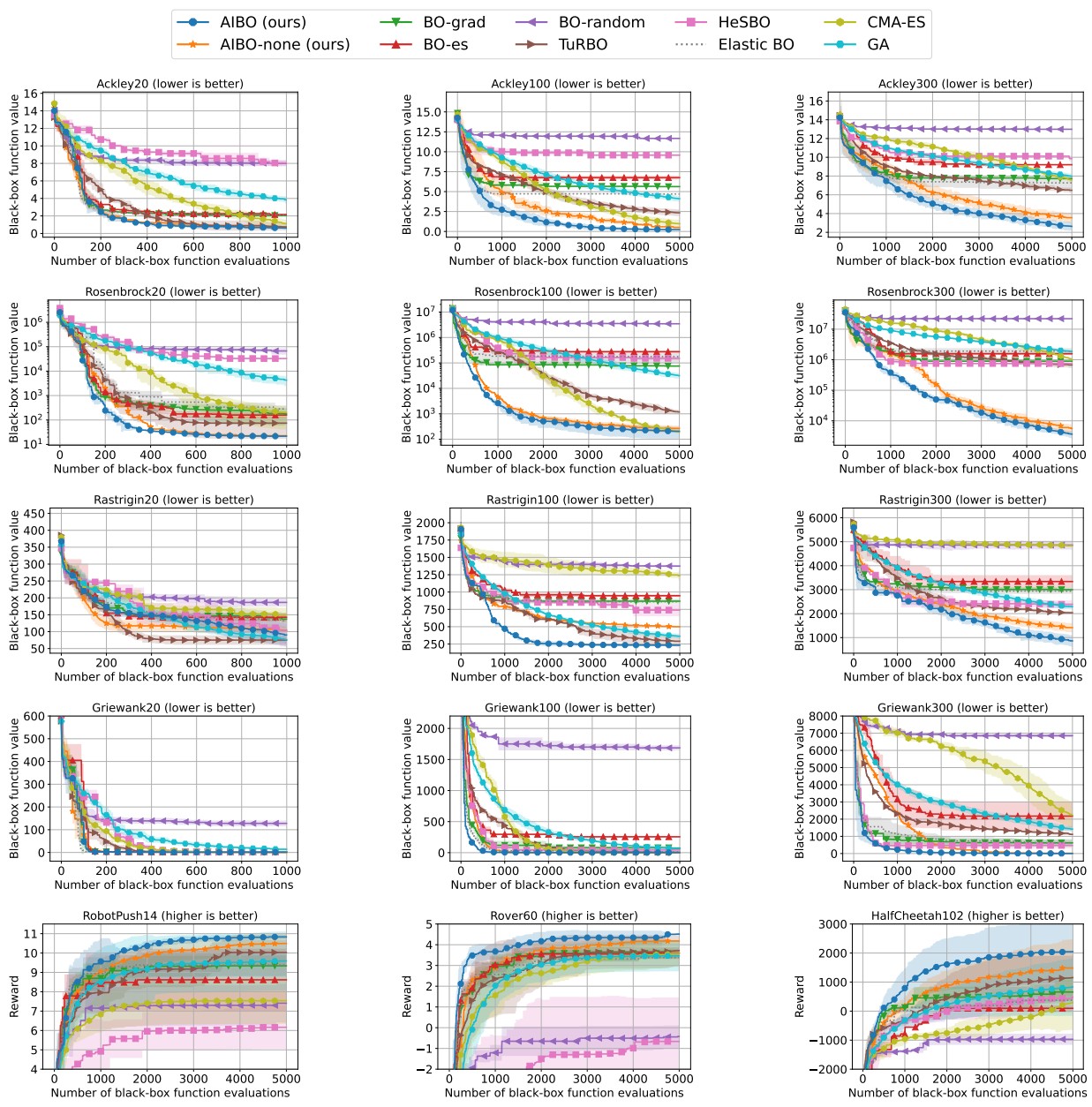

Figure 2: Results on both synthetic functions (lower is better) and real-world problems (higher is better). AIBO consistently improves BO-grad on all test cases and outperforms other competing baselines in most cases.

methods. For HeSBO, we use a target dimension of 8 for the 14D robot pushing task, 20 for the 102D robot locomotion task and 100D or 300D synthetic functions, and 10 for other tasks. Other settings are default in the reference implementations.

CMA-ES and GA are used to demonstrate the effectiveness of AF itself. Given that AIBO employs AF to further search the query point from the initial candidates generated by CMA-ES and GA black-box optimizers, if the AF is not sufficiently robust, the performance of AIBO might be inferior to CMA-ES/GA. For CMA-ES, the initial standard deviation is set to 0.2, and the rest of the parameters are defaulted in

pycma (Hansen et al., 2022). For GA, the population size is set to 50, and the rest of the parameters are defaulted in pymoo (Blank & Deb, 2020).

AIBO-none is a variant of AIBO. In each BO iteration, following the initialization of the AF maximization process, AIBO-none directly selects the point with the highest AF value while AIBO uses a gradient-based AF maximizer to further search points with higher AF values. This comparison aims to assess whether better AF maximization can improve performance.

### 6.1.2 Results

Fig.2 reports the full comparison results about the black-box function performance of our method AIBO with various baselines on all the benchmarks. We use UCB1.96 (UCB with $\beta_t = 1.96$) as the default AF.

**AIBO versus BO-grad**  While the performance varies across target functions, AIBO consistently improves BO-grad on all test cases. Especially for synthetic functions which allow flexible dimensions, AIBO shows clear advantages in higher dimensions (100D and 300D). We also observe that BO-grad exhibits a similar convergence rate to AIBO at the early search stage. This is because AF maximization is relatively easy to fulfil when the number of samples is small. However, as the search progresses, more samples can bring more local optimums to the AF, making the AF maximization process increasingly harder.

**AIBO versus CMA-ES/GA**  As AIBO introduces CMA-ES and GA black-box optimizers to provide initial points for AF maximization, comparing AIBO with CMA-ES and GA will show whether the AF is good enough to make the AF maximization process find better points than the initial points provided by CMA-ES/GA initialization. Results show AIBO outperforms CMA-ES and GA in most cases except for the 20D Rastrigin function, where GA shows superior performance. However, in the next section, we will demonstrate that adjusting UCB's beta from 1.96 to 1 will enable AIBO to maintain its performance advantage over GA. This suggests that with the appropriate choice of the AF, BO's model-based AF can offer a better mechanism for trading off exploration and exploitation compared to heuristic GA/CMA-ES algorithms.

**AIBO versus other HDBO methods**  When compared to representative HDBO methods, including TuRBO, Elastic BO and HeSBO, AIBO performs the best in most cases except for the 20D Rastrigin function, for which TuRBO shows the fastest convergence. However, for higher dimensions (100D and 300D), AIBO performs better than TuRBO on this function.

**AIBO versus AIBO-none**  Without the gradient-based AF optimizer, AIBO-none still shows worse performance than AIBO. This indicates that better AF maximization can improve the BO performance. This trend can also be observed in the results of standard BO with different AF maximizers, where BO-grad and BO-es outperform BO-random.

Overall, these experimental results highlight the importance of the AF maximization process for HDBO, as simply changing the initialization of the AF maximization process brings significant improvement.

## 6.2 Evaluation under Different AFs

We also evaluate the performance of AIBO and BO-grad under different AFs. Besides the default AF setting UCB1.96 (UCB with $\beta_t = 1.96$), we also select UCB1 ($\beta_t = 1$), UCB4 ($\beta_t = 4$) and EI as the AF, respectively. This aims to provide insights into how well AIBO enhances BO-grad across different AF settings, shedding light on its robustness and effectiveness across diverse contexts.

Fig. 3 shows a comprehensive evaluation of the effectiveness of our AIBO method across various AFs. Changing the AF has a noticeable impact on performance, highlighting the importance of AF selection. If an inappropriate AF is used, such as using UCB4 in Rastrigin20, the performance improvements achieved through the use of AIBO remain highly limited. Despite that, the results we obtained are highly encouraging. While different AFs exhibit varying convergence rates, we consistently observe a noteworthy enhancement in the performance of our method when compared to the standard BO-grad approach. The advantage is

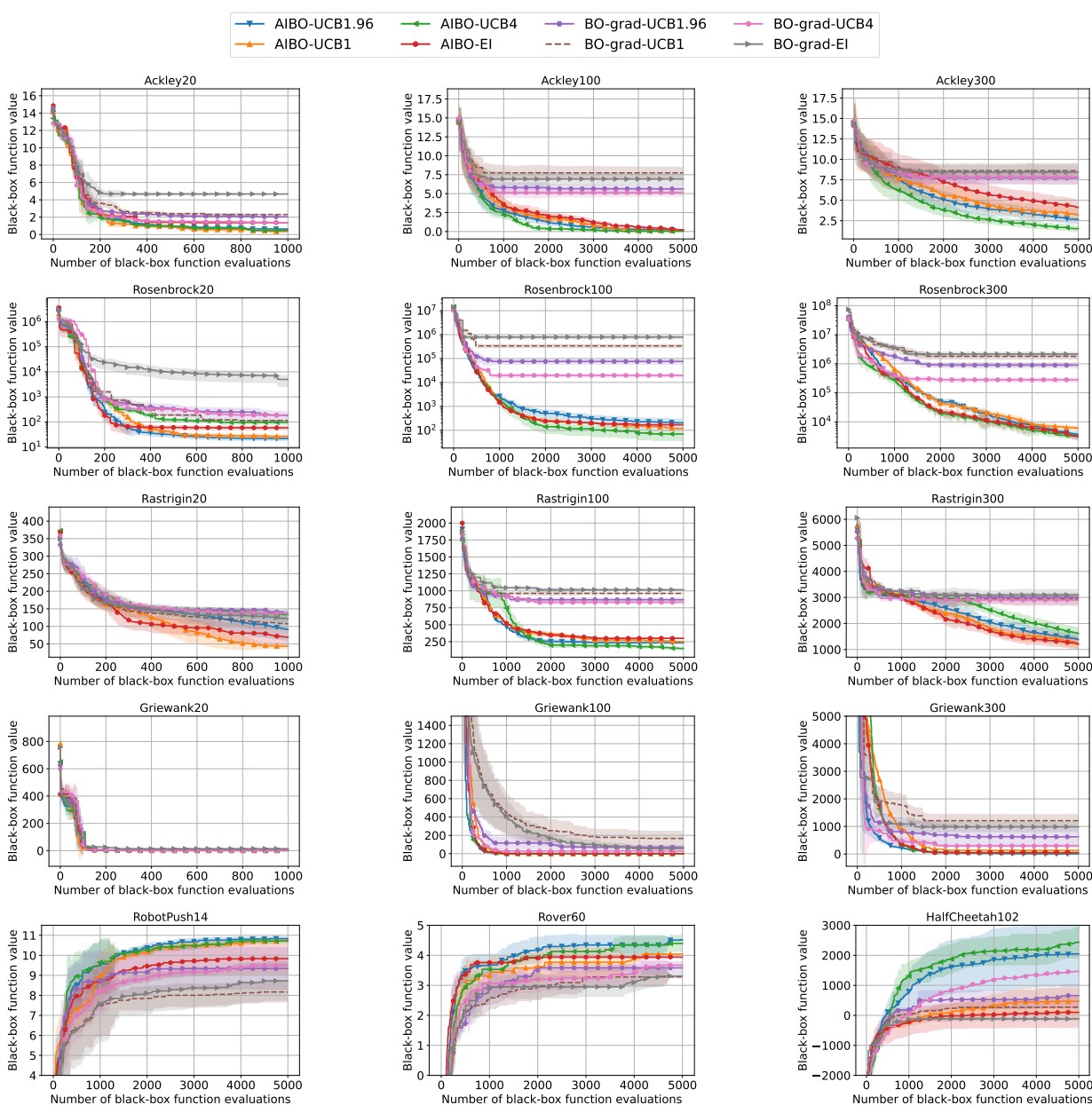

Figure 3: Evaluating the performance of AIBO and BO-grad under different AFs on both synthetic functions (lower is better) and real-world problems (higher is better).

clearer in higher dimensions (100D and 300D) than in lower dimensions (20D). These findings highlight the robustness and effectiveness of our initialization method across different AFs.

## 6.3 Over-exploration of Random Initialization

The aforementioned experimental results have demonstrated that heuristic AF maximizer initialization in AIBO leads to significant end-to-end BO performance improvements compared to random initialization. In this subsection, we evaluate AIBO's three initialization strategies in terms of AF values, GP posterior mean, and GP posterior variance under different AF settings.

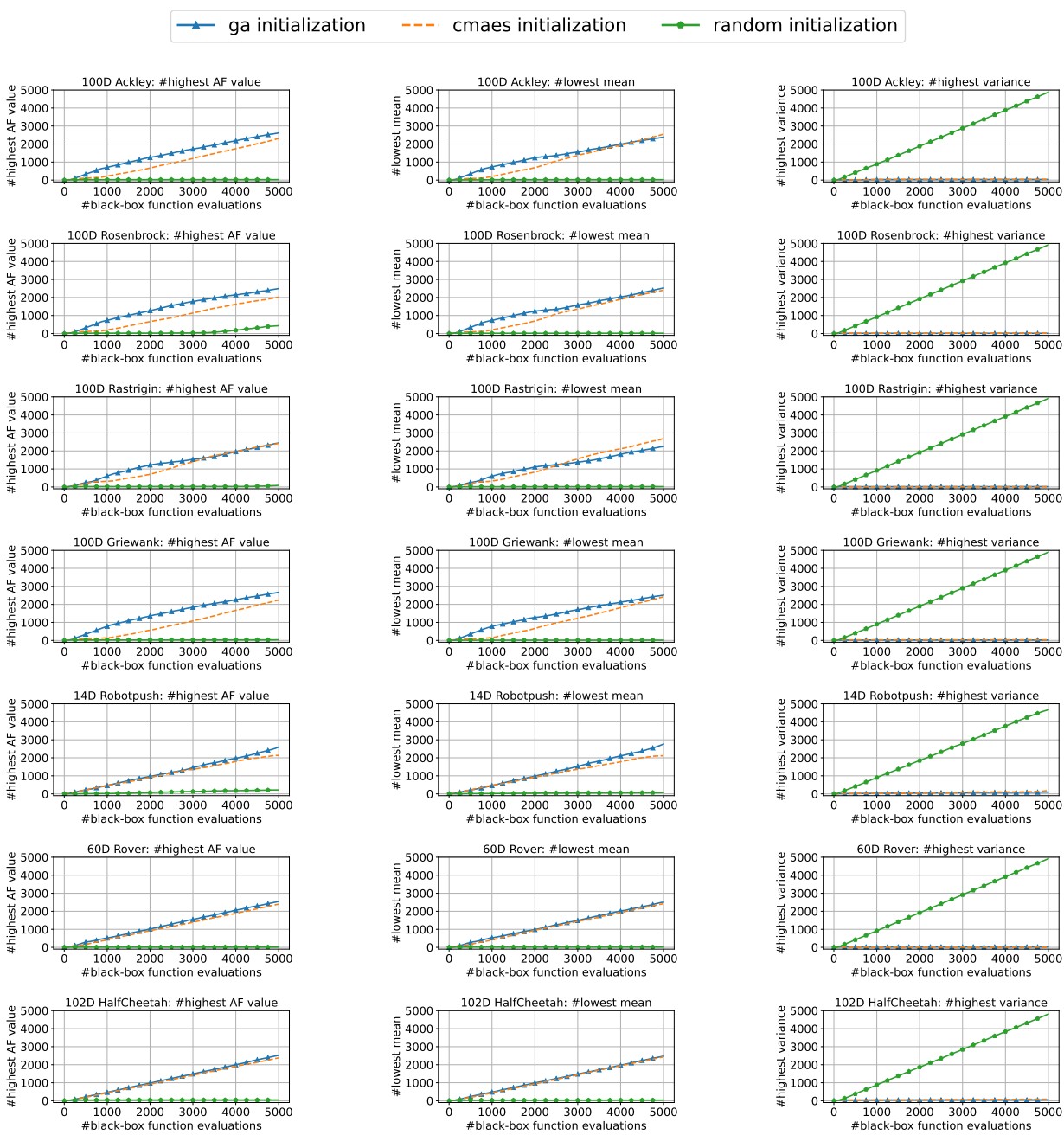

Figure 4: Evaluating AIBO's three initialization strategies in terms of AF values, GP posterior mean, and GP posterior variance when using UCB1.96 as the AF. The **left column** shows the number of times each initialization achieves the highest AF value among all the three strategies throughout the search process. Similarly, the **middle column** and **right column** indicate instances of achieving the lowest posterior mean (exploitation) and the highest posterior variance (exploration), respectively.

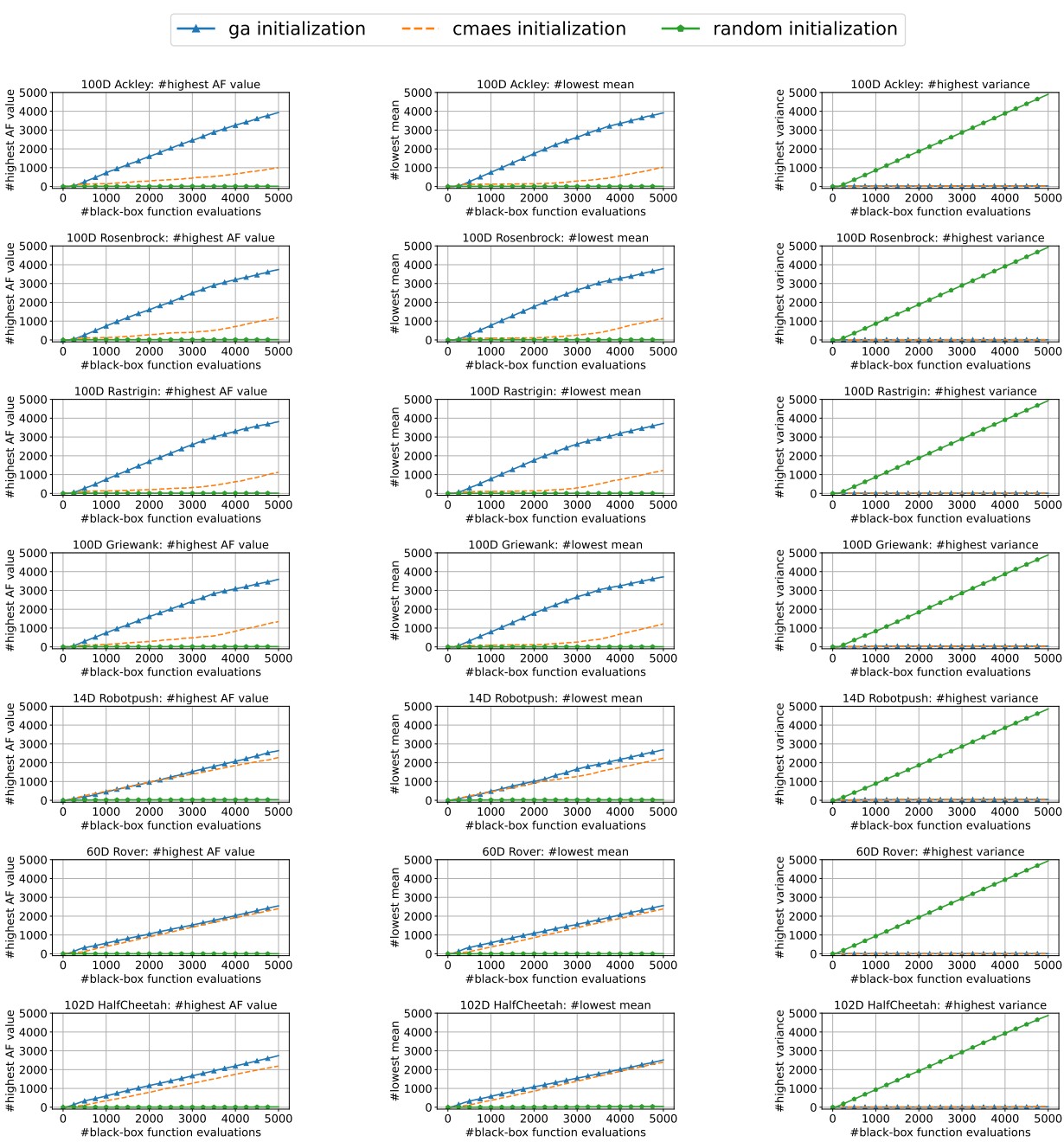

Figure 5: Evaluating AIBO's three initialization strategies in terms of AF values, GP posterior mean, and GP posterior variance when using UCB1 as the AF. The **left column** shows the number of times each initialization achieves the highest AF value among all the three strategies throughout the search process. Similarly, the **middle column** and **right column** indicate instances of achieving the lowest posterior mean (exploitation) and the highest posterior variance (exploration), respectively.

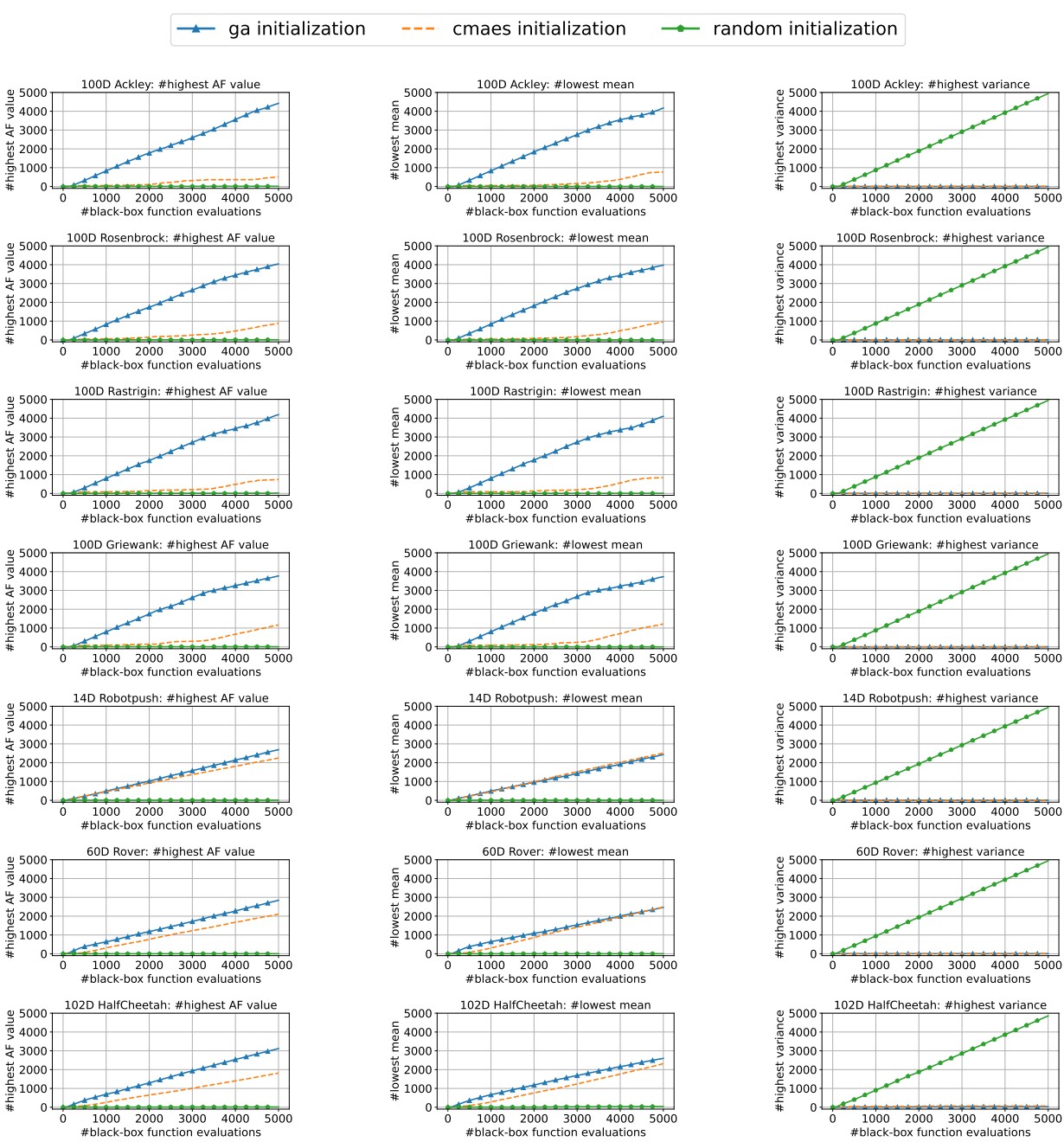

Figure 6: Evaluating AIBO's three initialization strategies in terms of AF values, GP posterior mean, and GP posterior variance when using EI as the AF. The **left column** shows the number of times each initialization achieves the highest AF value among all the three strategies throughout the search process. Similarly, the **middle column** and **right column** indicate instances of achieving the lowest posterior mean (exploitation) and the highest posterior variance (exploration), respectively.

In each iteration of AIBO, each initialization $o^i$ yields a candidate $x_t^i$ after AF maximization (Line 11 in Algorithm 1). For each initialization, we count the number of times $x_t^i$ achieves the highest AF value among $\{x_t^0, x_t^1, x_t^2\}$ until the current iteration. This number will show what initialization dominates the search process by yielding the highest AF value. Similarly, we also count the number of times $x_t^i$ achieves the lowest GP posterior mean (exploitation) and highest GP posterior variance (exploration), respectively. This will examine how different initialization schemes trade-off between exploration and exploitation.

The left column in Fig. 4 shows the number of times each initialization achieves the highest AF value among all the three strategies throughout the search process when using UCB1.96 ($\beta_t = 1.96$) as the AF. The middle and right columns indicate the number of times each initialization achieves the lowest posterior mean (exploitation) and the highest posterior variance (exploration), respectively. Compared to CMA-ES/GA initialization, random initialization always yields lower AF values and higher posterior variance, leading to over-exploration.

This over-exploration caused by random initialization is not exclusive to the UCB1.96 AF. As shown in Figs. 5 and 6, when decreasing $\beta_t$ from 1.96 to 1, or using EI as the AF, random initialization still yields lower AF values and higher posterior variance. This is due to the curse of the dimensionality. Since the search space size grows much faster than sampling budgets as the dimensionality increases, most regions are likely to have a high posterior variance. Given that more samples can bring more local optimums to AFs as the search progresses, random initialization tends to guide the AF maximizer to find local optimums in regions of high posterior variance. Even if the AF is designed to prioritize regions with lower GP posterior mean values for exploitation (e.g. UCB with a lower $\beta_t$), these regions are sparse and may be inaccessible through random initialization. AIBO is designed to mitigate the drawback of random initialization, and the results presented here validate AIBO indeed achieves better AF maximization by optimizing the initialization phase.

### 6.4 Ablation Study

To better understand the role played by each initialization strategy in AIBO, we evaluate the three individual initialization strategies in AIBO, leading to three variants of AIBO: AIBO_ga, AIBO_cmaes and AIBO_random. We note that AIBO_random is equivalent to BO-grad discussed earlier. Our fourth variant, AIBO_gacma, removes the random initialization strategy in AIBO.

As shown in Fig. 7, advanced heuristic initialization strategies like GA and CMA-ES show better performance than random initialization in most cases, showing the advantage of a heuristic algorithm over random initialization. Using a single advanced heuristic initialization, AIBO_ga and AIBO_cmaes achieve similar performance to AIBO in most cases. This suggests that CMA-ES and GA can be the main source of performance improvement for AIBO. AIBO_gacma shows a similar performance to AIBO in all cases. This is because GA/CMA-ES initialization dominates AIBO's search process.

Besides, although AIBO_cmaes is competitive in most problems, it is ineffective for the 14D robot pushing problem, suggesting there is no "one-fits-for-all" heuristic across tasks. By incorporating multiple heuristics, the ensemble strategy used by AIBO gives a more robust performance than the individual components.

### 6.5 Comparison with Other Initialization Strategies

In previous experiments, we implemented random initialization by selecting the top-n points with the highest AF values as the initial points from a large set of random points. Some existing BO implementations have employed alternative initialization strategies. The impact of these methods has not been systematically evaluated. We conduct a comparison of the following methods alongside AIBO: BO-cmaes_grad, BO-boltzmann_grad and BO-Gaussian_grad.

#### 6.5.1 Setup

BO-cmaes_grad uses CMA-ES to optimize the AF to provide better initial points for further gradient-based AF maximization. We note that in this case, CMA-ES is used directly for AF optimization. In contrast, the "CMA-ES" in AIBO is used to provide initial points by leveraging the history of black-box optimization.

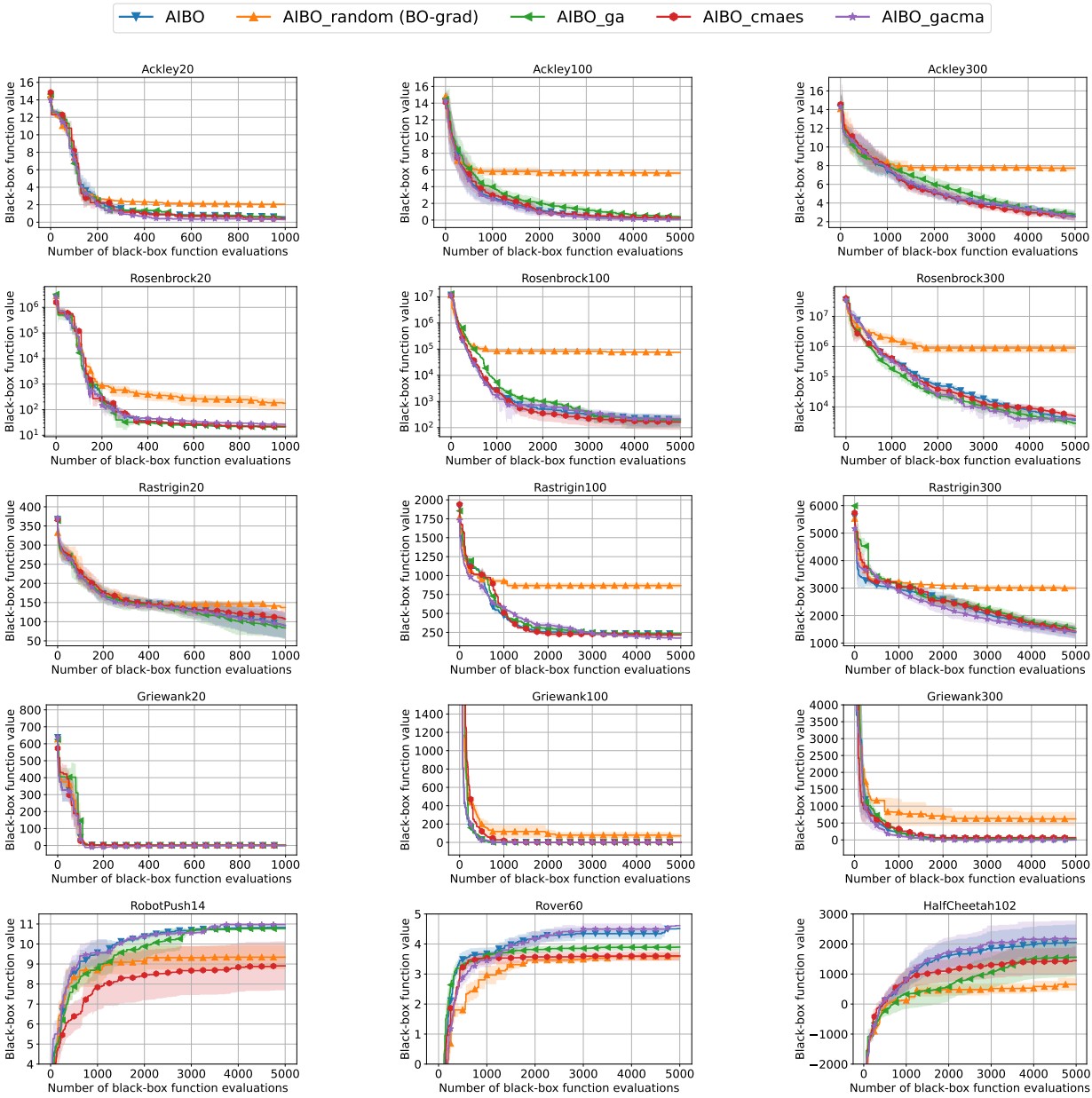

Figure 7: Comparing AIBO to its variants AIBO_gacma, AIBO_ga, AIBO_cmaes and AIBO_random (BO-grad) . While a single advanced heuristic heuristic strategy CMA-ES/GA already performs well in most cases, using the ensemble strategy improves the robustness.

Comparing these two methods will reveal the importance of the black-box optimization history in the AF maximization process.

BO-boltzmann_grad refers to the default implementation in BoTorch, which uses Boltzmann sampling to generate initial points for the gradient-based AF maximization. In each BO iteration, it evaluates the AF on a large set of points and then uses an annealing heuristic (rather than top-n) to select the restart points.

BO-Gaussian_grad uses a Gaussian spray around the incumbent best to generate initial points for the gradient-based AF maximizer. This initialization strategy has been used in Spearmint (Snoek et al., 2012), and we replace the AF maximizer with the advanced gradient-based method for a fair comparison.

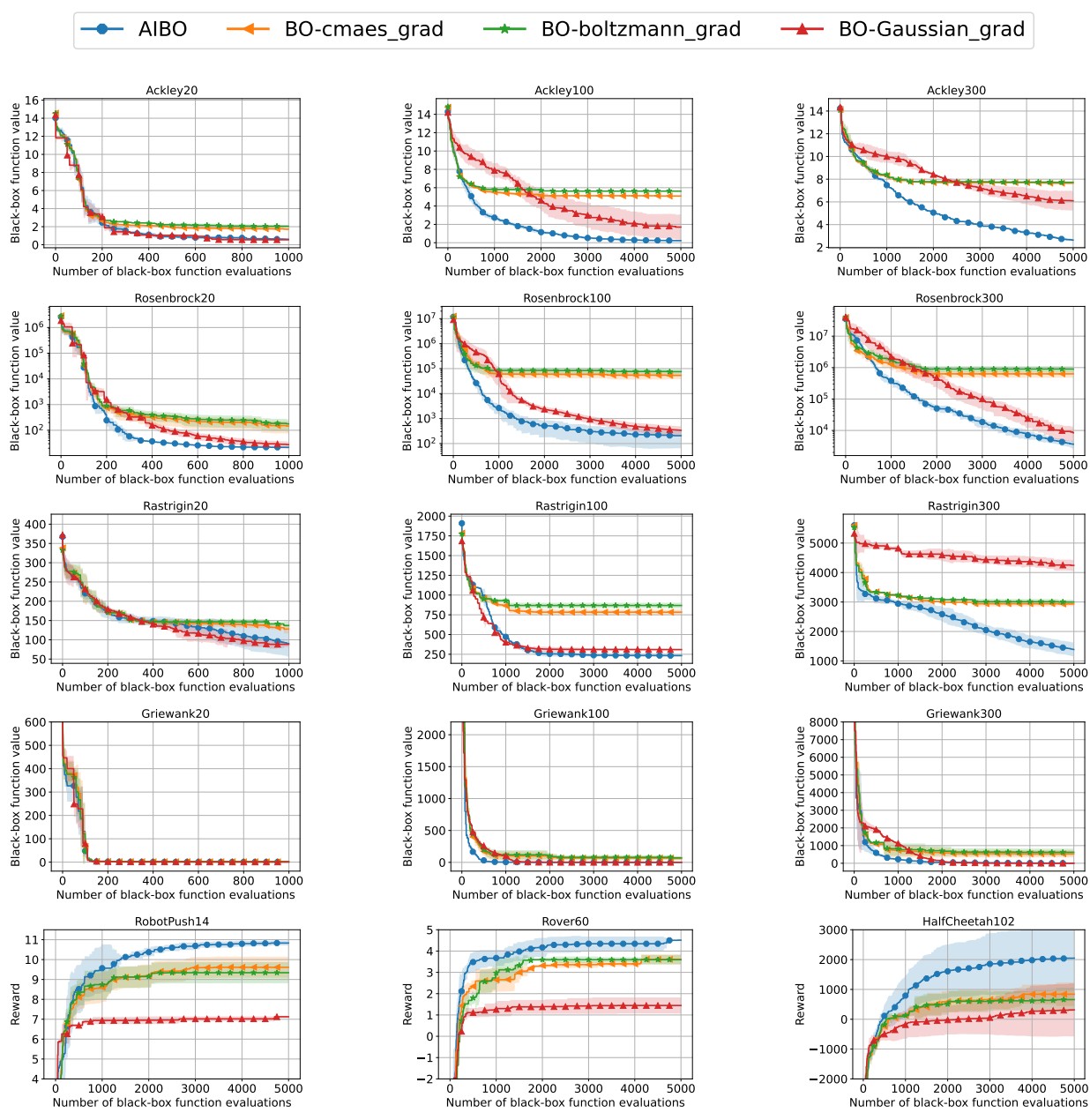

Figure 8: Comparing AIBO to standard BO with other AF initialization methods that do not use random search on both synthetic functions (lower is better) and real-world problems (higher is better).

### 6.5.2 Results

Fig. 8 presents the comparison result between AIBO and other initialization strategies. BO-cmaes_grad and BO-boltzmann_grad exhibit significantly inferior performance compared to AIBO. Both approaches do not leverage prior black-box optimization history and instead attempt to optimize the AF in the global space directly to provide initial points for further gradient-based AF optimization. This underscores the challenges of AF optimization in high-dimensional problems and the importance of utilizing the black-box optimization history. BO-Gaussian_grad takes into account the best points from the past black-box optimization history as a basis for maximizing the AF. This approach performs well in some cases (e.g., Rastrigin100) but may lead to a significant performance drop in other situations (e.g., Robotpush14) due to over-exploitation. Overall, AIBO exhibits significantly better performance compared to these non-random initialization strategies.

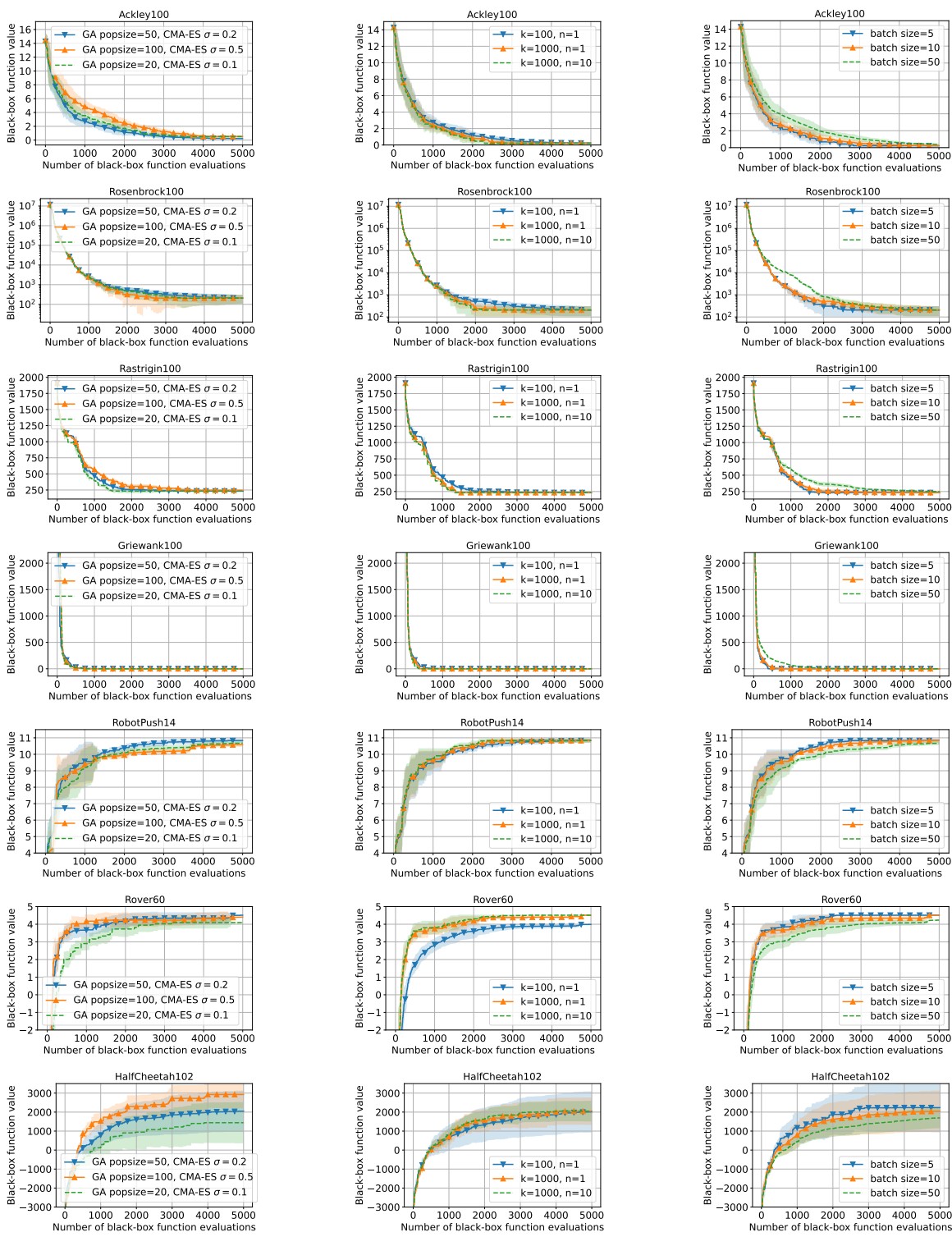

Figure 9: The impact of hyper-parameters on AIBO. The **left column** shows the impact of GA population size and CMA-ES initial standard deviation $\sigma$. The **middle column** reports the impact of the number of raw candidates generated from heuristics $k$ and the number of selected initial points $n$. The **right column** shows the impact of the batch size.

Table 2: Algorithmic runtime

|  | Synthetic | | | RobotPush | Rover | HalfCheetah |
|---|---|---|---|---|---|---|
| Dimensions | 20 | 100 | 300 | 14 | 60 | 102 |
| #Samples | 1000 | 5000 | 5000 | 5000 | 5000 | 5000 |
| AIBO | 8 min | 2.5 h | 3.6 h | 1.8 h | 2 h | 2.5 h |
| BO-grad | 12 min | 3.3 h | 5 h | 2.5 h | 3 h | 4 h |

## 6.6 Evaluation under Different Hyper-Parameters

Multiple hyper-parameters in AIBO, including GA population size, CMA-ES initial standard deviation $\sigma$, the number of raw candidates generated from heuristics $k$, the number of selected initial points $n$, and the batch size could impact its performance.

**GA pop size and CMA-ES $\sigma$** As AIBO employ heuristics to initialize the AF maximization process, these heuristics' hyper-parameters control the quality of initial points of the AF maximization process and affect the trade-off between exploration and exploitation. A larger GA population size and a larger CMA-ES initial standard deviation will encourage more exploration. As shown in the left column of Fig. 9, different tasks favour different trade-offs. A more exploratory setting (popsize=100 and $\sigma = 0.5$) works well for the HalfCheetah102 task but reduces the performance on Ackley100. Our experiments suggest opting for a GA population size between 20 and 100 and selecting the CMA-ES initial $\sigma$ value within the range of 0.2 to 0.5.

**$k$ and $n$** Based on Algorithm 1, increasing the number of raw candidates generated from heuristics $k$ and the number of selected initial points $n$ might help AF maximization but requires more calculation. However, as shown in the middle column of Fig. 9, increasing $k$ and $n$ does not yield significant performance improvement in most cases except for the Rover60 task. We recommend setting $k$ to $100 \sim 1000$ and setting $n$ to $1 \sim 10$.

**Batch size** As shown in the right column of Fig. 9, AIBO performs well across different batch sizes, and reducing the batch size can slightly enhance convergence speed in all cases.

## 6.7 Algorithmic Runtime

In Table 2, we provide the algorithmic running time (excluding the time spent evaluating the objective function) for our method with a batch size of 10. For comparison, we also show the algorithmic runtime of BO-grad. The experiments are run on an NVIDIA RTX 3090 GPU equipped with a 20-core Intel Xeon Gold 5218R CPU Processor. As described in Sec 6.1.1, to show the effectiveness of AIBO, BO-grad is allowed to perform more costly AF maximization. AIBO uses less algorithmic runtime because it costs less AF maximization time than the standard BO-grad method. AIBO's algorithmic runtime is also acceptable for actual expensive black-box optimization tasks (only several hours for a few thousand evaluations).

## 7 Conclusion and Future Work

We have presented a large-scale empirical study to understand the impact of the acquisition function (AF) maximizer initialization process when applying Bayesian optimization (BO) for high-dimensional problems. Our extensive experiments show that the AF maximizer initialization can greatly impact the realization of the AF, and the widely random initialization strategy may fail to unlock the potential of an AF.

We then propose AIBO, a framework to optimize the initialization phase of AF maximization BO. AIBO is designed to overcome the limitation of the random initialization technique for high-dimensional BO. AIBO employs a simple yet effective optimization strategy. It employs multiple heuristic optimizers to generate the raw samples for the acquisition function maximizer to better trade-off exploration and exploitation.

We evaluate AIBO by applying it to synthetic test functions, robot control, and planning tasks. Experimental results show that AIBO can significantly boost the standard BO's performance in high-dimensional problems and outperform prior high-dimensional BO techniques.

Previous theoretical studies on the convergence of BO methods have largely relied on a fundamental assumption that the AF could be globally optimized. While this assumption may hold in low-dimensional tasks, our large-scale empirical study suggests that this assumption could be violated in high-dimensional problems. We show that how the AF is optimized plays a crucial role in determining the overall efficacy of the BO method. Our future work will examine the theoretical underpinnings governing BO convergence when AF maximization is suboptimal. We hope this study, with extensive empirical evidence, can promote more research in high-dimensional BO by jointly optimizing the AF and the process of maximizing the AF.

## Acknowledgments

This work was supported in part by the UK Engineering and Physical Sciences Research Council (EPSRC) under grant agreement EP/X018202/1. For any correspondence, please contact Zheng Wang (Email: z.wang5@leeds.ac.uk).

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

# A   Appendix

## A.1   The case of over-exploitation

To show AIBO's random initialization can help alleviate the over-exploitation issue, we create a variant of our technique, AIBO_gacma, by removing random initialization. Using default hyperparameters, AIBO_gacma performs well in the RobotPush14 optimization task. However, after adjusting the hyperparameters to an over-exploitation case by setting GA population size to 3 and CMA-ES initial standard deviation to 0.01, we observe that AIBO_gacma performs less effectively. Upon reintroducing random initialization, we observed significant performance improvement, suggesting that random initialization could help mitigate over-exploitation.

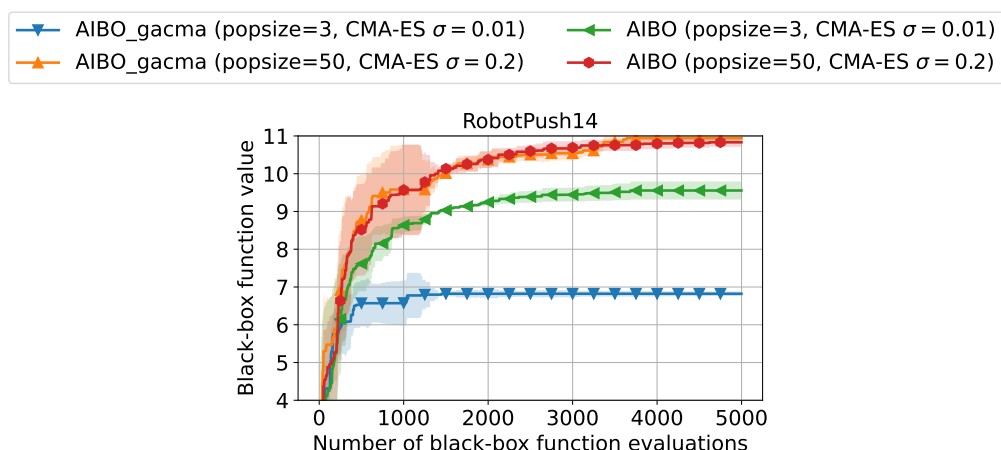

Figure 10: Comparision of AIBO and AIBO_gacma under a standard hyperparameter setting and a over-exploitative setting. Results show that random initialization could help mitigate over-exploitation.

## A.2   Details of GA and CMA-ES in AIBO

In this section, we explain how GA and CMA-ES optimizers in AIBO are initialized (Line 3 in Algorithm 1), updated (Line 16 in Algorithm 1) and asked to generate raw candidates (Line 8 in Algorithm 1). We use the implementations in pycma and pymoo for the CMA-ES and the GA strategies, respectively.

**Initialization of GA**   At the beginning of the search process in AIBO, we will draw $N$ samples uniformly and evaluate them to produce GA's initial population.

**Candidate generation of GA**   To generate new candidates, GA will sequentially perform selection, crossover and mutation operations. The selection operation aims to select individuals from the current population of GA to participate in mating (crossover and mutation). The crossover operation combines parents into one or several offspring. Finally, the mutation operation generates the final candidates based on the offspring created through the crossover. It helps increase the diversity in the population. The selection, crossover and mutation operations used in AIBO are shown as follows.

- *Tournament Selection*: It involves randomly picking $T$ individuals from the population, comparing their fitness, and selecting the individual with the highest fitness. This process is repeated to fill the new generation. We use the default setting of pymoo, i.e., $T = 2$.

- *Simulated Binary Crossover (SBX)*: This is a widely used crossover technique. A binary notation can represent real values, and then point crossovers can be performed. SBX simulated this operation by using an exponential probability distribution simulating the binary crossover. For this operation, we also use the default SBX implementation of pymoo, where the crossover probability is set to 0.5.

- *Polynomial Mutation*: This mutation follows the same probability distribution as the simulated binary crossover to introduce small, random changes to individuals in the population to maintain genetic diversity. We use the default polynomial mutation implementation of pymoo, where the mutation probability is set to 0.9.

**Update of GA**  We update the population of GA using the most recently evaluated samples in AIBO. Here, we sort the samples based on their fitness (black-box function value), ultimately retaining those with superior fitness.

**Initialization of CMA-ES**  The key of CMA-ES is a multivariate normal distribution $\mathcal{N}(m_k, \sigma_k^2 C_k)$, which is initialized at the beginning of the search process in AIBO. In particular, we will draw $N$ samples uniformly and evaluate them. The coordinates of the sample with the best black-box function value will used as the initial mean vector $m_0$. The step size $\sigma_k$ is initialized to a constant $\sigma_0 = 0.2$, and the covariance matrix $C_k$ is initialized as an identity matrix $C_0 = I$.

**Candidate generation of CMA-ES**  At each iteration $k$, CMA-ES generates candidates by sampling from its current multivariate normal distribution, i.e.,

$$
\begin{aligned}
x_i &\sim \mathcal{N}(m_k, \sigma_k^2 C_k) \\
&\sim m_k + \sigma_k \times \mathcal{N}(0, C_k)
\end{aligned}
\tag{2}
$$

**Update of CMA-ES**  The update of CMA-ES involves updating the multivariate normal distribution $\mathcal{N}(m_k, \sigma_k^2 C_k)$. Assuming the batch size of AIBO is $\lambda$, the samples $x_i$ are evaluated on the objective function $f$ to be minimized. The only feedback from the objective function here is ordering the sampled candidate solutions due to the indices $i : \lambda$. Denoting the $f$-sorted candidate solutions as

$$
\{x_{i:\lambda} \mid i = 1 \dots \lambda\} = \{x_i \mid i = 1 \dots \lambda\} \text{ and } f(x_{1:\lambda}) \leq \cdots \leq f(x_{\mu:\lambda}) \leq f(x_{\mu+1:\lambda}) \leq \cdots,
$$

the new mean value is computed as

$$
\begin{aligned}
m_{k+1} &= \sum_{i=1}^{\mu} w_i \, x_{i:\lambda} \\
&= m_k + \sum_{i=1}^{\mu} w_i \, (x_{i:\lambda} - m_k)
\end{aligned}
$$

where the positive (recombination) weights $w_1 \geq w_2 \geq \cdots \geq w_\mu > 0$ sum to one. Typically, $\mu \leq \lambda/2$ and the weights are chosen such that $\mu_w := 1/\sum_{i=1}^{\mu} w_i^2 \approx \lambda/4$.

The step-size $\sigma_k$ is updated using "cumulative step-size adaptation" (CSA), sometimes also denoted as "path length control". The evolution path (or search path) $p_\sigma$ is updated first.

$$
p_\sigma \leftarrow \underbrace{(1 - c_\sigma)}_{\text{discount factor}} p_\sigma + \overbrace{\sqrt{1 - (1 - c_\sigma)^2}}^{\text{complements for discounted variance}} \underbrace{\sqrt{\mu_w} \, C_k^{-1/2} \overbrace{\frac{m_{k+1} - m_k}{\sigma_k}}^{\text{displacement of } m}}_{\text{distributed as } \mathcal{N}(0, I) \text{ under neutral selection}}
$$

$$
\sigma_{k+1} = \sigma_k \times \exp\left( \frac{c_\sigma}{d_\sigma} \underbrace{\left( \frac{\|p_\sigma\|}{\mathrm{E}\,\|\mathcal{N}(0, I)\|} - 1 \right)}_{\text{unbiased about 0 under neutral selection}} \right)
$$

where

- $c_\sigma^{-1} \approx n/3$ is the backward time horizon for the evolution path $p_\sigma$ and larger than one ($c_\sigma \ll 1$ is reminiscent of an exponential decay constant as $(1 - c_\sigma)^k \approx \exp(-c_\sigma k)$ where $c_\sigma^{-1}$ is the associated lifetime and $c_\sigma^{-1} \ln(2) \approx 0.7 c_\sigma^{-1}$ the half-life),

- $\mu_w = \left( \sum_{i=1}^{\mu} w_i^2 \right)^{-1}$ is the variance effective selection mass and $1 \le \mu_w \le \mu$ by definition of $w_i$,

- $C_k^{-1/2} = \sqrt{C_k}^{-1} = \sqrt{C_k^{-1}}$ is the unique symmetric square root of the inverse of $C_k$, and

- $d_\sigma$ is the damping parameter usually close to one. For $d_\sigma = \infty$ or $c_\sigma = 0$, the step size remains unchanged.

The step size $\sigma_k$ is increased if and only if $\|p_\sigma\|$ is larger than the expected value

$$
\mathrm{E}\,\|\mathcal{N}(0, I)\| = \sqrt{2}\,\Gamma\left(\frac{n+1}{2}\right) / \Gamma\left(\frac{n}{2}\right)
$$

$$
\approx \sqrt{n}\left(1 - \frac{1}{4n} + \frac{1}{21n^2}\right)
$$

and decreased if it is smaller. For this reason, the step-size update tends to make consecutive steps $C_k^{-1}$-conjugate, in that after the adaptation has been successful $\left(\frac{m_{k+2} - m_{k+1}}{\sigma_{k+1}}\right)^T C_k^{-1} \frac{m_{k+1} - m_k}{\sigma_k} \approx 0$.

Finally, the covariance matrix is updated, where again, the respective evolution path is updated first.

$$
p_c \leftarrow (1 - c_c)p_c + \mathbf{1}_{[0, \alpha\sqrt{n}]}(\|p_\sigma\|)\sqrt{1 - (1 - c_c)^2}\sqrt{\mu_w}\frac{m_{k+1} - m_k}{\sigma_k}
$$

$$
C_{k+1} = (1 - c_1 - c_\mu + c_s)C_k + c_1 p_c p_c^T + c_\mu \sum_{i=1}^{\mu} w_i \frac{x_{i:\lambda} - m_k}{\sigma_k}\left(\frac{x_{i:\lambda} - m_k}{\sigma_k}\right)^T
$$

where $T$ denotes the transpose and

- $c_c^{-1} \approx n/4$ is the backward time horizon for the evolution path $p_c$ and larger than one,

- $\alpha \approx 1.5$ and the indicator function $\mathbf{1}_{[0, \alpha\sqrt{n}]}(\|p_\sigma\|)$ evaluates to one iff $\|p_\sigma\| \in [0, \alpha\sqrt{n}]$ or, in other words, $\|p_\sigma\| \le \alpha\sqrt{n}$, which is usually the case,

- $c_s = (1 - \mathbf{1}_{[0, \alpha\sqrt{n}]}(\|p_\sigma\|)^2)c_1 c_c(2 - c_c)$ makes partly up for the small variance loss in case the indicator is zero,

- $c_1 \approx 2/n^2$ is the learning rate for the rank-one update of the covariance matrix and

- $c_\mu \approx \mu_w/n^2$ is the learning rate for the rank-$\mu$ update of the covariance matrix and must not exceed $1 - c_1$.

### A.3 Additional real-world benchmarks

Here we provide two additional real-world benchmarks Lasso-DNA (Šehić et al., 2021) and Nasbench (Ying et al., 2019), to improve the claim for the real-world performance of our method. The Lasso-DNA task minimizes the mean squared error (MSE) of weighted Lasso sparse regression for real-world DNA datasets by tuning 180 parameters. The Nasbench task is to design a neural architecture cell topology defined by a DAG with 7 nodes and up to 9 edges to maximize the CIFAR-10 test-set accuracy, subject to a constraint where the training time was less than 30 minutes. We follow the parameterization approach used by Letham et al. (2020) to make the Nasbench task become a 36-dimensional black-box function.

For Lasso-DNA, AIBO still significantly outperforms BO-grad. For Nasbench-101, AIBO only shows a modestly improved performance compared to BO-grad. This is because the search space size of Nasbench-101 is small (only 423,624 samples). For such a small search space size, AF maximization is no longer a challenge.

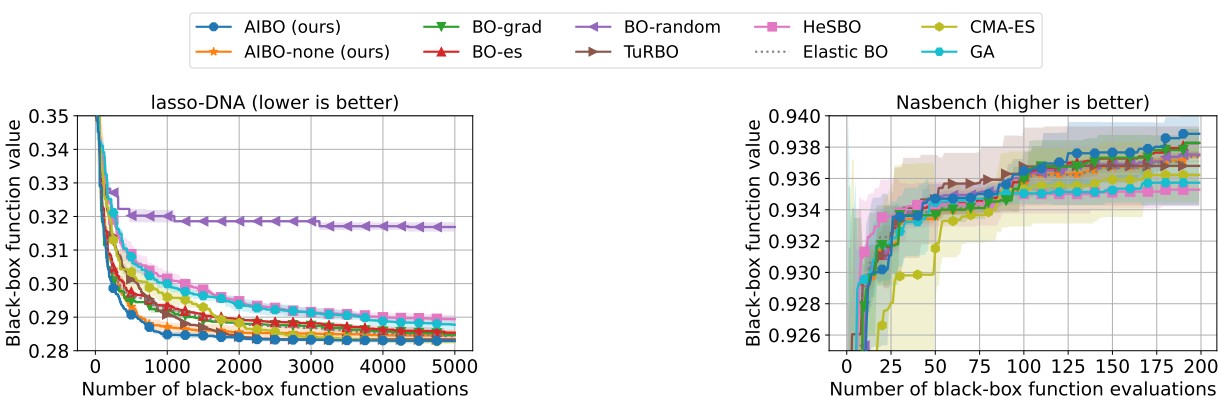

Figure 11: Results on two additional real-world problems Lasso-DNA and NasBench.

### A.4 Comparision with DIRECT initialization

As the DIRECT (Jones et al., 1993) algorithm is widely used for the acquisition function optimization, we also use it to optimize the AF for generating initial points for further gradient-based optimizers. The corresponding BO method is denoted as BO-DIRECT_grad. We compare BO-DIRECT_grad with AIBO in all the benchmarks. As shown in Fig. 12, AIBO outperforms BO-DIRECT_grad in all cases, and the performance gap is notable in most cases.

### A.5 Monte Carlo acquisition function

Here, we provide details about how Monte Carlo acquisition functions are calculated. Many common acquisition functions can be expressed as the expectation of some real-valued function of the model output(s) at the design point(s):

$$\alpha(X) = \mathbb{E}\big[a(\xi) \mid \xi \sim \mathbb{P}(f(X) \mid \mathcal{D})\big]$$

where $X = (x_1, \ldots, x_q)$, and $\mathbb{P}(f(X) \mid \mathcal{D})$ is the posterior distribution of the function $f$ at $X$ given the data $\mathcal{D}$ observed so far.

Evaluating the acquisition function thus requires evaluating an integral over the posterior distribution. In most cases, this is analytically intractable. In particular, analytic expressions generally do not exist for batch acquisition functions that consider multiple design points jointly (i.e. $q > 1$).

An alternative is to use (quasi-) Monte-Carlo sampling to approximate the integrals. An Monte-Carlo (MC) approximation of $\alpha$ at $X$ using $N$ MC samples is

$$\alpha(X) \approx \frac{1}{N} \sum_{i=1}^{N} a(\xi_i)$$

where $\xi_i \sim \mathbb{P}(f(X) \mid \mathcal{D})$.

For instance, for MC-estimated Expected Improvement, we have:

$$\mathrm{qEI}(X) \approx \frac{1}{N} \sum_{i=1}^{N} \max_{j=1,\ldots,q} \big\{\max(\xi_{ij} - f^*, 0)\big\}, \qquad \xi_i \sim \mathbb{P}(f(X) \mid \mathcal{D})$$

where $f^*$ is the best function value observed so far (assuming noiseless observations).

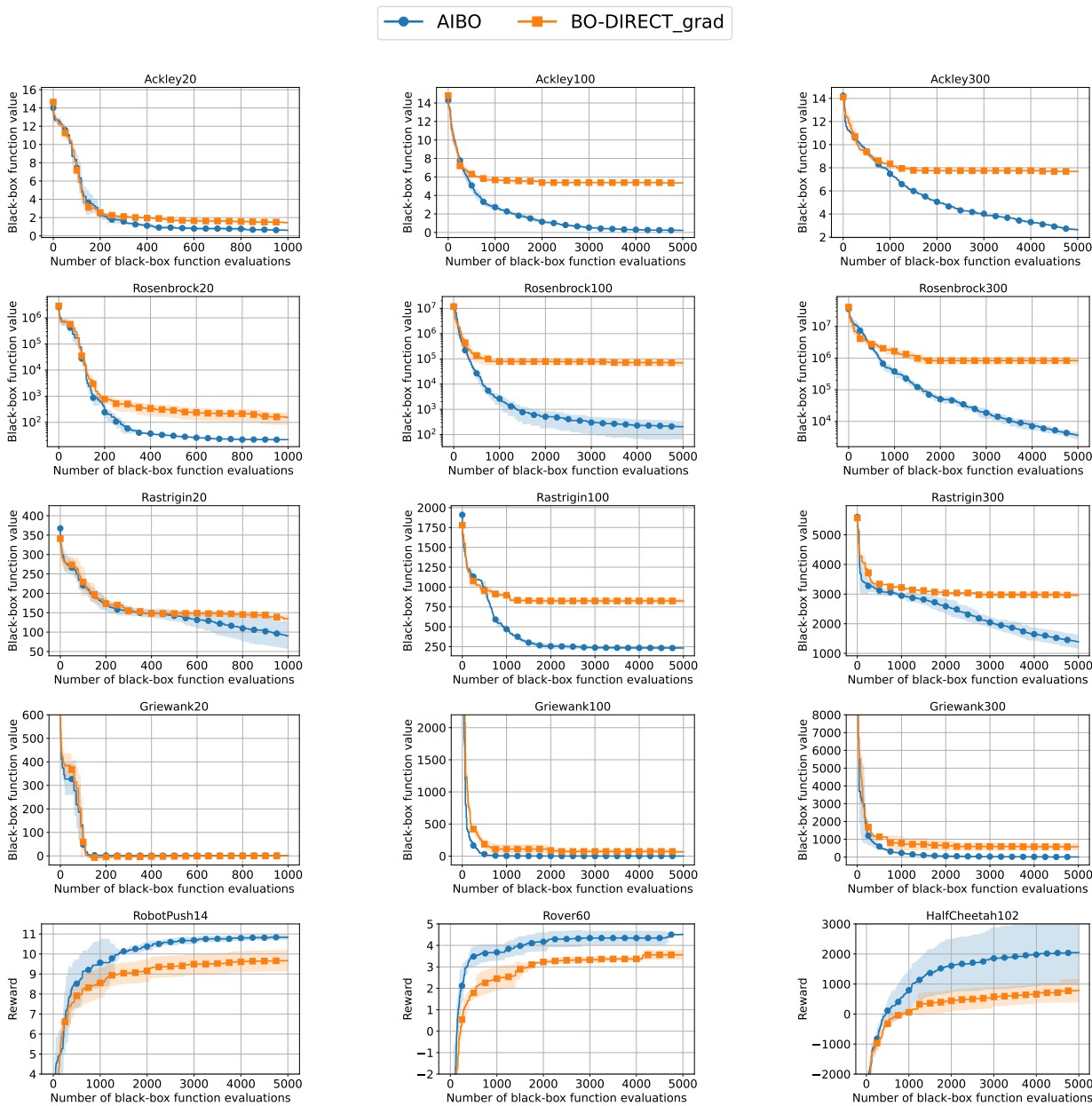

Figure 12: Comparing AIBO to BO-DIRECT_grad. AIBO outperforms BO-DIRECT_grad in all cases.

Using the reparameterization trick (Wilson et al., 2018),

$$\text{qEI}(X) \approx \frac{1}{N} \sum_{i=1}^{N} \max_{j=1,\ldots,q} \left\{ \max\left( \mu(X)\_j + (L(X)\epsilon_i)\_j - f^*, 0 \right) \right\}, \qquad \epsilon_i \sim \mathcal{N}(0, I)$$

where $\mu(X)$ is the posterior mean of $f$ at $X$, and $L(X)L(X)^T = \Sigma(X)$ is a root decomposition of the posterior covariance matrix.

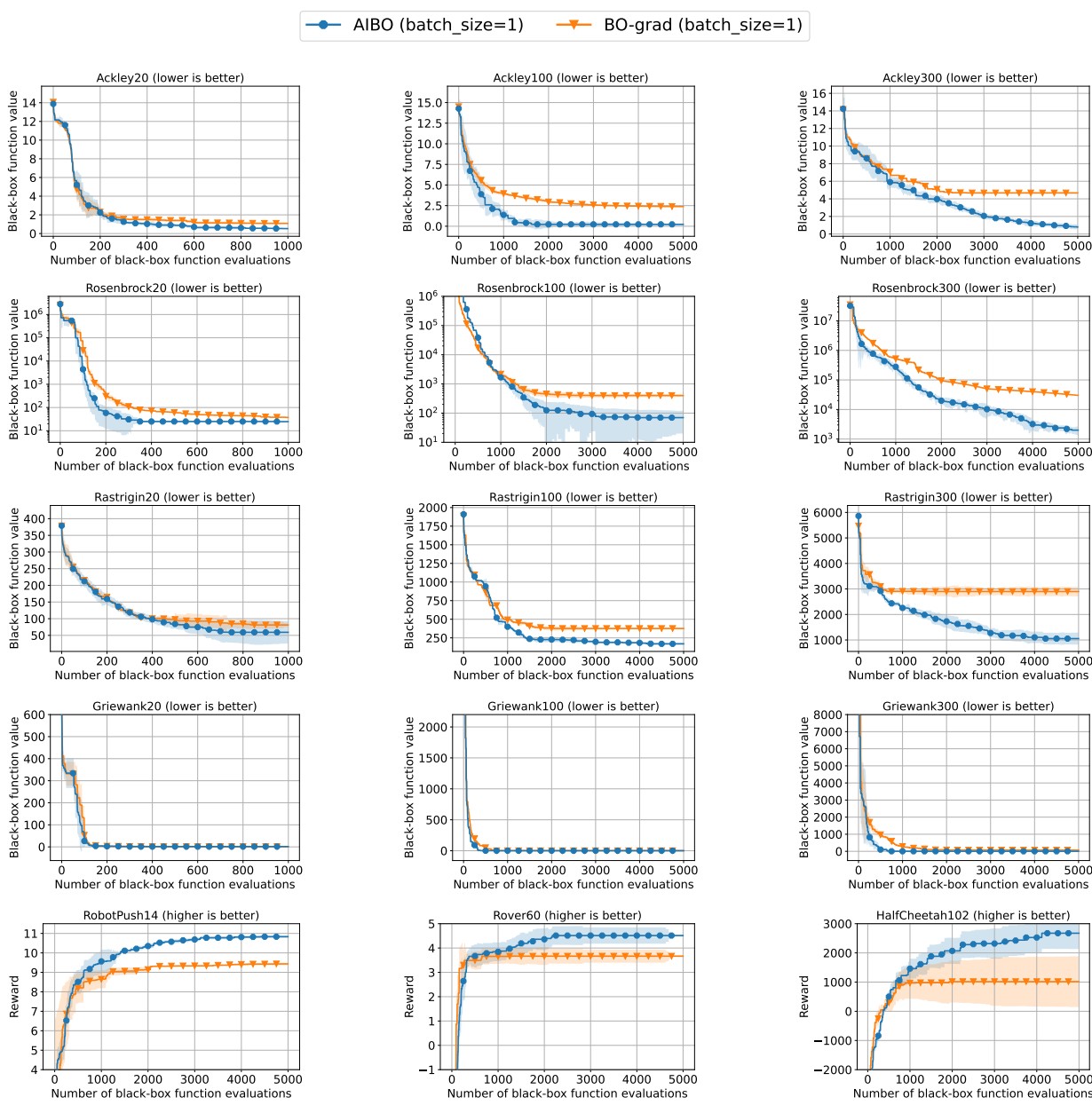

Figure 13: Evaluating AIBO and BO-grad on analytical UCB AF with $\beta_t = 1.96$. We set the batch size to 1 to support the analytical AF. AIBO still outperforms BO-grad in all cases.

## A.6 Evaluations on analytical acquisition functions

To evaluate our method on analytical AFs (unsuitable for batch BO), we set the batch size to 1. As shown in Fig. 13 Using analytical UCB AF $\beta_t = 1.96$, AIBO still outperforms BO-grad in all cases.

## A.7 The impact of Acquisition Function on the diversity of GA population

Our heuristic initialization process is AF-related, as it depends on past samples selected by the given AF. Usually, a more explorative AF setting will make the heuristic initialization in AIBO also more explorative.

For instance, if the AF formula leans towards exploration in GA, the GA population composed of samples chosen by this AF will have greater diversity, generating more diverse raw candidates.

To illustrate this point. We select two different AF settings: UCB1.96 ($\beta_t = 1.96$), and UCB9 ($\beta_t = 9$, more exploratory). When applying AIBO to Ackley100, we calculate the average distance between individuals in the GA population at each BO iteration, which is a good measure of the population diversity. We repeated this experiment 50 times. As shown in Fig. 14, AIBO with UCB9 achieves more diverse GA populations than AIBO with UCB 1.96, suggesting that a more explorative AF setting will make GA initialization more explorative.

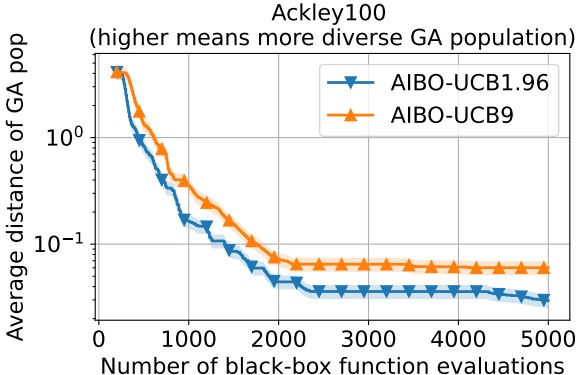

Figure 14: Evaluating the average distance between individuals in the GA population when applying AIBO-UCB1.96 and AIBO-UCB9 (more exploratory) to the Ackley100 function. With the same number of BO iterations, AIBO-UCB9 always owns a more diverse GA population than AIBO-UCB1.96, suggesting that a more exploratory AF setting will make the GA initialization in AIBO also more exploratory.

