# OpenReview forum: "Unleashing the Potential of Acquisition Functions in High-Dimensional Bayesian Optimization"
_TMLR — Accepted by TMLR_

### Review · Reviewer_YXGR · 2023-11-08

**Summary Of Contributions:**

This work consists of a comprehensive empirical study of the impact of **Acquisition Function (AF) maximizer initialization** in Bayesian optimization.

The authors present enough empirical evidence (via simulated and real data) to argue that the commonly used random initialization strategy does not often exploit the full potential of the Bayesian optimization procedure.

In fact, the authors conclude that random AF maximizer initialization tends to over-explore, and that simply increasing the number of random initialization restarts does not resolve the issue.

Motivated by these insights, the authors present a framework (consisting of a combination and selection among existing heuristics) for a better initialization of the AF maximizer. They propose to combine $k$ points per-initialization strategy, with a later selection of the top-$n$ candidates among them. The aim is to provide a better and more diverse set of points for the AF maximizer to start from, for improved downstream BO performance.

This work's experimental evidence showcases how the proposed ---heuristic-based combine-then-select--- initialization framework can, across a variety of synthetic and real-world applications, outperform each of its components in isolation, as well as other state-of-the-art methods.

**Audience:**

Yes

**Broader Impact Concerns:**

This work does not discuss broader impact concerns.

**Claims And Evidence:**

Yes

**Requested Changes:**

The authors argue for the need for other initialization strategies as a countermeasure for the over-exploration of the commonly used random initialization.

However, one may wonder whether such exploration is actually a desirable feature in BO, so that one fully covers and evaluates the domain of interest (with an obvious increase in computational cost, of course). On the contrary, the objective of this work seems to focus on exploiting as much as possible.

Hence, I would encourage the authors to discuss in detail how important this trade-off is:
   - e.g., whether there might be other cases (with a more complex optimization landscape) in which AIBO might exploit too early without fully exploring the full domain, or whether this will never occur (as long as random initialization is part of AIBO).
   - how to reconcile or use model-based AF's within AIBO that the authors acknowledge _"can offer a better mechanism for trading off exploration and exploitation compared to heuristic GA/CMA-ES algorithms"_.

In addition, section 6.6 provides an evaluation of hyperparameter tuning in a single example (Ackley100). The contribution of this work would be strengthen with a more comprehensive hyperparameter evaluation across datasets and a discussion on how it impacts BO performance across tasks.

Finally, some theoretical motivation or discussion would significantly strengthen the impact of this work.

These are other minor suggestions:
   - The Introduction section already covers and discusses certain concepts and works cited in the Background section, making the latter a bit redundant. I would suggest the authors consider merging these 2 sections into a single introduction one, where all the necessary background is discussed as needed.
   - Section 4.2, last sentence of first paragraph seems to be incomplete: _"GPyTorch, the overhead of running BO with a GP model for a few thousand evaluations would be acceptable"_
   - Section 5.2, typo in first paragraph: _"In Section 6.2, We evaluate"_ -> _"In Section 6.2, we evaluate"_
   - Section 6.3, typo in second paragraph: _"highest GP posterior variance exploration)"_ -> _"highest GP posterior variance (exploration)"_

**Strengths And Weaknesses:**

The main strength of this work is the thorough experimental evaluation provided, with

   - comparison to state-of-the-art baselines
   - an ablation study to understand the benefits and drawbacks of each heuristic

The conclusions of interest drawn from the presented evaluation are:
   - the impact of AF maximizer initialization is not negligible in BO
   - random initialization tends to explore more than heuristic alternatives
   - selecting initial points from a pool of several heuristic-based candidates can improve downstream BO performance

The main limitations of this work are:
   - The studied problem (i.e., the impact of AF maximizer initialization) is very specific and narrow
   - The proposed algorithm is limited to a "combine-then-select'' approach: i.e., combine several heuristic initializations to propose $k$ points each, then to select the top $n$ candidates
   - There is a lack of theoretical motivation, justification or analysis on why the proposed framework may or may not work

---

> ### Author Response · Authors · 2023-12-06
> **Response to Reviewer YXGR**
>
> We thank the reviewer for the insightful feedback.
>
> - *“The proposed algorithm is limited to a "combine-then-select'' approach: i.e., combine several heuristic initializations to propose points each, then to select the top candidates”*
>
> Yes, but we found this simple strategy is also effective. It also allows us to examine if GA/CMA-ES initialization could find better AF values than random initialization. Our scheme only chooses GA/CMA-ES initialization if it yields larger AF values than random initialization. This is now clarified in Sec. 4.1.
>
> - *“whether there might be other cases (with a more complex optimization landscape) in which AIBO might exploit too early without fully exploring the full domain, or whether this will never occur (as long as random initialization is part of AIBO)”*
>
> It is possible that AIBO can exploit too early. However, this can be mitigated by tuning the hyperparameters of the heuristic initialization strategies. Random initialization can also help alleviate this issue. This is now explained in Sec. 4.1.
>
> We have added new experiments in Appendix A.1 to illustrate this point. In this experiment, we create a new variant of our technique, AIBO_gacma, by removing random initialization. Using default hyperparameters, AIBO_gacma performs well in the RobotPush14 optimization task. However, after adjusting the hyperparameters to an over-exploitation case by setting GA population size to 3 and CMA-ES initial standard deviation to 0.01, we observe that AIBO_gacma performs less effectively. Upon reintroducing random initialization, we observed significant performance improvement, suggesting that random initialization could help mitigate over-exploitation.
>
> - *“how to reconcile or use model-based AF's within AIBO that the authors acknowledge "can offer a better mechanism for trading off exploration and exploitation compared to heuristic GA/CMA-ES algorithms”*
>
> The trade-off between exploration and exploitation is jointly determined by the model-based AF and the initialization strategy. In each BO iteration, while the AF maximizer is initialized by heuristic strategies, these heuristic optimizers are also updated with observations selected by the AF. This enables the AF and heuristic optimizers to collaborate, enhancing the robustness of AF maximization. For example, if an AF is inclined towards exploration,  then the population of GA composed of samples selected by this AF will have better diversity (i.e., a more explorative AF will make the GA initialization in AIBO also more explorative). We have clarified these in Sec. 4.1.
>
>
>
>
> - *“The contribution of this work would be strengthen with a more comprehensive hyperparameter evaluation across datasets and a discussion on how it impacts BO performance across tasks.”*
>
> Thank you for the suggestion. We have added a dedicated hyperparameter evaluation and discussion to Sec. 6.6.
>
> - *“Some theoretical discussion would significantly strengthen the impact of this work.”*
>
> Our paper aims to provide a comprehensive and empirical study of the impact of AF maximizer initialization on high-dimensional BO. Many previous studies have theoretically demonstrated the convergence of BO methods. Yet, these theoretical analyses were based on the assumption that the AF could be globally optimized. Instead, our work empirically demonstrates that this assumption does not hold in high-dimensional situations, and moreover, the optimization process of the AF significantly impacts the final performance of the BO.
>
> We agree with the reviewer that a theoretical understanding would be very valuable, and we hope that the empirical observations made in this work will encourage more theoretical investigation in this direction. This is now discussed in Sec. 7.
>
> - *“These are other minor suggestions”*
>
> We have made the suggested changes. Thank you.

---

> > ### Comment · Reviewer_YXGR · 2023-12-20
> > **Thank you for your detailed responses and the updated manuscript**
> >
> > Dear Authors,
> >
> > I appreciate your detailed response to my (and other reviewers') comments.
> >
> > I believe the revised manuscript is more complete, with clarifications that help the reader.
> >
> > Thank you!

---

### Review · Reviewer_4uty · 2023-11-09

**Summary Of Contributions:**

In this paper, the authors study the problem of improving initializations of acquisition function optimizations in Bayesian optimization algorithms. The main contribution is a novel initialization scheme that employs three different methods (CMA-ES, GA, and random) to propose initialization points based on already queried points. The authors provided extensive empirical evaluations on synthetic and real-world optimization tasks to show the efficacy of the proposed method. Different acquisition functions are used to show the generality as well.

**Audience:**

Yes

**Broader Impact Concerns:**

No concerns.

**Claims And Evidence:**

Yes

**Requested Changes:**

1. I found the description of the method incomplete. Specifically, AIBO uses CMA-ES and GA to provide initial point candidates. However, details are lacking. For example, how does one update the mean vector $m$ and the covariance matrix $C$ after acquiring new samples? For GA, how does one perform the mutation and crossover operations? In fact, there are different versions of GA so it is important to specify those details.


2. In Section 6.4, given that AIBO_ga and AIBO_cmaes are both competitive with the full version AIBO. I think it makes sense to add AIBO_{ga+cmaes} as well to the comparison.


3. In Section 6.7 algorithmic runtime comparison, I did not understand why AIBO would be faster than BO-grad. AIBO requires additional function calls for GA and CMA-ES. Is the speedup related to software implementation or the underlying hardware?


4. I would consider running NAS-Bench-101 or a comparable benchmark to improve the claim for real-world performance.

**Strengths And Weaknesses:**

Strengths:
1. Improving acquisition function optimization is an important problem within Bayesian optimization that has received relatively little attention. This work will likely be relevant to many researchers in the BO community.

2. The proposed method is easy to understand and implement. The empirical results are mostly convincing in the general superiority over existing initialization methods. As a result, we could potentially see this method as the new default in acquisition function optimization.

3. The authors provided a comprehensive list of experiments to analyze various aspects of the proposed method. They include varying the acquisition function, varying hyperparameters in acquisition functions, and ablation study for individual components in the method. These experiments illustrate the importance of including all three methods in the initialization process.

Weaknesses:
1. Some parts of the submission could use more details explaining the practical implementations. Please see the requested changes section.

2. The real-world tasks are still relatively artificial. The experiment on HalfCheetah shows large variances with 50 trials, which makes it difficult to interpret the results. There are more realistic tasks such as NAS-Bench-101 for optimizing neural network architectures worthy of consideration.

---

> ### Author Response · Authors · 2023-12-06
> **Response to Reviewer 4uty**
>
> We thank the reviewer for the constructive feedback.
>
> - *Details of GA and CMA-ES in AIBO*
>
> We use the implementations in pycma (https://github.com/CMA-ES/pycma) and pymoo (https://github.com/anyoptimization/pymoo) for the CMA-ES and the GA initialization strategies, respectively. We have provided more details in Appendix A.2.
>
> - *“In Section 6.4, given that AIBO_ga and AIBO_cmaes are both competitive with the full version AIBO. I think it makes sense to add AIBO_{ga+cmaes} as well to the comparison.”*
>
> Agree. We have added the results of AIBO_gacma in Sec. 6.4. It shows a similar performance to AIBO in all cases.
>
> - *“In Section 6.7 algorithmic runtime comparison, I did not understand why AIBO would be faster than BO-grad. AIBO requires additional function calls for GA and CMA-ES. Is the speedup related to software implementation or the underlying hardware?”*
>
> As described in Sec 6.1.1, to show the effectiveness of AIBO, BO-grad is allowed to perform more costly AF maximization. The additional overhead of function calls for GA and CMA-ES is negligible compared to the AF maximization time. This is now clarified in Sec 6.7.
>
> - *“I would consider running NAS-Bench-101 or a comparable benchmark to improve the claim for real-world performance.”*
>
> We have added two additional real-world benchmarks Lasso-DNA [1] and NasBench-101, in Appendix A.3.  For Lasso-DNA, AIBO still outperforms BO-grad. For Nasbench-101, AIBO shows a modestly improved performance than BO-grad. This is because the search space size of Nasbench-101 is small (only 423,624 samples). For such a small search space size, AF maximization is no longer a challenge. This is also clarified in Appendix A.3.
>
>
> [1] Šehić, Kenan, et al. "LassoBench: A high-dimensional hyperparameter optimization benchmark suite for lasso." International Conference on Automated Machine Learning. PMLR, 2022.

---

### Review · Reviewer_GeKQ · 2023-11-23

**Summary Of Contributions:**

The paper considers initial value settings of the acquisition function maximization in Bayesian optimization (BO). The basic idea is to use multiple heuristic optimizer such as CMAES and GA and optimize the acquisition function from each of generated initial values. The performance is evaluated several functions with a variety of settings.

**Audience:**

Yes

**Claims And Evidence:**

Yes

**Requested Changes:**

Justification of the tuning strategy of heuristic optimizers using past data should be clarified. For example, in the case of covariance of CMAES, the directions that has never been explored may have zero covariance, which may inhibit the exploration to unknown directions. Although the past data base update of heuristic optimizers is seemingly one of main claims, I'm not fully convinced by a rationale behind this concept.

I think an appropriate initialization should depend on the acquisition function. If the acquisition function tends to exploit, a strategy based on past information may be advantageous for the acquisition function maximization, while if the acquisition function is more explorative, past information may hinder the efficient maximization. For example, in the case of UCB, this balance is changed by beta. How superiority of the single initialization strategy for different exploit-exploration acquisition functions is justified?

Some related approaches are missed to discuss:
- DIRECT is widely used for the acquisition function optimization, though it is not mentioned. It should also be able to use for generating initial points for gradient-based optimizers.
- LineBO (ICML2019) is another well-known high dimensional BO, for which the authors should have mentioned.

How is the MC acquisition function calculated? Explanation in page 6 (Batch Bayesian optimization) is unclear. It should be described in more detail.

MC based acquisition functions may have largely different behavior from analytical acquisition functions (such as the standard formulation of EI). For example, analytical acquisition functions typically only depends on posterior mean and variance, by which the acquisition function surface becomes flat in the region that posterior mean and variance unchange, which may not happen in MC based EI and UCB because the values are fluctuated by random sampling, and it may have effect on the behavior of the gradient-based optimizers. Discussion in the paper is only for MC acquisition function? If so, it should be clearly explained. If not, evaluations on analytical acquisition functions would be desired.

**Strengths And Weaknesses:**

S: As mentioned in the paper, the topic has not been widely studied, and may have potential importance for practical performance of BO.

S: Comprehensive experimental evaluation is performed.

W: Justification of the proposed method is not fully clear to me.

W: Some related work should be mentioned.

---

> ### Author Response · Authors · 2023-12-06
> **Response to Reviewer GeKQ**
>
> We thank the reviewer for the valuable feedback.
>
> - *“Justification of the tuning strategy of heuristic optimizers using past data should be clarified. For example, in the case of covariance of CMAES, the directions that has never been explored may have zero covariance, which may inhibit the exploration to unknown directions. ”*
>
> This is a great point. We have added more details about how heuristic optimizers generate candidates and update themselves based on past data in Appendix A.2.
>
> For CMA-ES, the covariance matrix CMA-ES is initialized at the beginning of the BO search, and each direction (dimension) will be assigned an initial covariance. AIBO's default setting configures this initial covariance to be 0.2, ensuring exploration across all directions and preventing any situation where a dimension remains unexplored with a covariance value of zero. This is now explained at Sec. 4.1.
>
> - *“I think an appropriate initialization should depend on the acquisition function … while if the acquisition function is more explorative, past information may hinder the efficient maximization. How superiority of the single initialization strategy for different exploit-exploration acquisition functions is justified?”*
>
> This is an interesting point, which we discussed as follows.
>
> We agree that the initialization can depend on the underlying Acquisition Function (AF). Our heuristic initialization process is actually AF-related, as it depends on past samples which are selected by the given AF. Usually, a more explorative AF setting will make the heuristic initialization in AIBO also more explorative. For instance, in GA, if the AF formula leans towards exploration, the GA population composed of samples chosen by this AF will have greater diversity, leading to generating more diverse raw candidates.
>
> We have added new experiments in Appendix A.7 to illustrate this point. We select two different AF settings: UCB1.96 (beta=1.96), and UCB9 (beta=9, more explorative). When applying AIBO to the Ackley100 function, at each BO iteration, we calculate the average distance between individuals in the GA population, which is a good measure of the population diversity. We observe that AIBO with UCB9 always achieves more diverse GA populations than AIBO with UCB 1.96, suggesting that a more explorative AF setting will make GA initialization more explorative.
>
> Furthermore,  our goal is not to maximize any arbitrary AF (e.g. a pure explorative AF) but to unlock the potential of reasonable and effective AF settings to enhance end-to-end BO performance. The heuristic initialization strategies may not always improve the optimization of nearly pure explorative AFs, e.g., UCB with beta=10000. Instead, we find that for reasonably designed AFs, heuristic initialization often finds better AF values than random initialization. The experimental results in Sec. 6.2 and Sec. 6.3 support this, showing that random initialization tends to over-explore and find poorer AF values for different but reasonable AF settings.
>
> Finally, heuristic initialization is not merely exploitative; it also incorporates exploratory mechanisms, and altering its hyperparameters can achieve different trade-offs. For example, increasing GA's population size or mutation probability actually enhances exploration. AIBO allows the use of various initialization strategies, including exploration-oriented random initialization, and permits adjustments to the hyperparameters of heuristic initialization methods, making it suitable for different tasks and corresponding different AFs. Results in Sec. 6.6 show that tuning the hyperparameters of heuristic initializations can enhance performance. Moreover, due to the robustness of heuristic algorithms, using a fixed hyperparameter setting still yields much better results than random initialization across different tasks.
>
> These are now clarified at Sec. 1 and Sec. 4.1.
>
> - *“Some related approaches are missed to discuss”*
>
> We have added the results of combining DIRECT and gradient-based optimizers for AF maximization in Appendix A.4. We also added the description of LineBO in Sec. 2. Thanks for the suggestions.
>
> - *Details of MC acquisition functions*
>
> We calculate the MC acquisition function (AF) by using quasi-Monte-Carlo sampling to approximate integrals over the posterior distribution that are required by analytic AFs.  The details are now explained in Appendix A.5.
>
> - *“If not, evaluations on analytical acquisition functions would be desired.”*
>
> To evaluate our method on analytical AFs (not suitable for batch BO), we set the batch size to 1. The corresponding results have been added in Appendix A.6. AIBO still outperforms BO-grad in all cases.

---

### Decision · Action_Editor_wwjR · 2023-12-22

**Recommendation:** Accept with minor revision

**Comment:**

The reviewers have raised different weaknesses and issues for the initial submission. After rebuttal, the authors have successfully addressed most concerns and requested changes. In the official recommendation, two reviewers vote to clearly accept this work and one reviewer leans toward acceptance.

I read the paper in detail and also believe this work is valuable for the BO community. One thing that can be improved is a clear and more accurate description of the existing initialization strategies. The current submission simply claims "random initialization is a typical default strategy for initializing the AF maximizer" and "this is the case for widely-used packages like BoTorch, skort, GPyOpt, and GPflowOpt". However, the initialization strategies (even if all belong to random research) are not the same for different packages. For example, according to Appendix B.1 in [1], GPyOpt "augments random restarts with the best points observed so far, or alternatively points generated via
Thompson sampling", and BoTorch "selects initial points by performing Boltzmann sampling on a set of random points according to their acquisition function value". More detailed descriptions of different initialization strategies are also provided there. Since this work focuses on the initialization for BO, a clear and accurate discussion on the initialization strategies for different packages/works could be beneficial to the reader. Therefore, I recommend **accept with minor revision** for this work.

In the revised paper, I expect the authors to:

+ Add a more comprehensive and detailed discussion on the initialization strategies for different BO methods/packages.

+ Double-check and ensure all the discussions/analyses provided in the rebuttal are well incorporated in the revised paper.

[1] Sebastian Ament, Samuel Daulton, David Eriksson, Maximilian Balandat, and Eytan Bakshy. Unexpected Improvements to Expected Improvement for Bayesian Optimization. NeurIPS 2023.

**Audience:**

All reviewers believe some individuals in TMLR's audience could be interested in the findings of this paper.

**Claims And Evidence:**

This work conducts a thorough empirical study on the effect of acquisition function initialization for high-dimensional Bayesian optimization (BO). The key finding is that the widely used random initialization strategy could perform poorly for high-dimensional problems. Based on this observation, this work then proposes an ensemble initialization strategy with multiple heuristic optimizers (CMA-ES, Genetic Algorithm, and random search) to generate better initial solutions. Comprehensive experiments on synthetic test functions and real-world applications show that the proposed ensemble initialization method can significantly improve BO's performance on high-dimensional problems.

The reviewers find the initialization strategy is important for Bayesian optimization but has not been widely studied, and the findings reported in this work could be valuable for the researchers/practitioners in the BO community. All reviewers believe the claims made in this work are well supported by the comprehensive experimental studies and analysis, and I totally agree.